# Occupancy Prediction in IoT-Enabled Smart Buildings: Technologies, Methods, and Future Directions

**DOI:** 10.3390/s24113276

**Published:** 2024-05-21

**Authors:** Irfanullah Khan, Ouarda Zedadra, Antonio Guerrieri, Giandomenico Spezzano

**Affiliations:** 1ICAR-CNR, Institute for High Performance Computing and Networking, National Research Council of Italy, Via P. Bucci 8/9C, 87036 Rende, Italy; giandomenico.spezzano@icar.cnr.it; 2DIMES Department, University of Calabria, Via P. Bucci, 87036 Rende, Italy; 3LabSTIC Laboratory, Department of Computer Science, 8 Mai 1945 University, P.O. Box 401, Guelma 24000, Algeria; zedadra.ouarda@univ-guelma.dz

**Keywords:** internet of things, occupancy detection, estimation and prediction, artificial intelligence, machine learning, smart buildings, cognitive buildings

## Abstract

In today’s world, a significant amount of global energy is used in buildings. Unfortunately, a lot of this energy is wasted, because electrical appliances are not used properly or efficiently. One way to reduce this waste is by detecting, learning, and predicting when people are present in buildings. To do this, buildings need to become “smart” and “cognitive” and use modern technologies to sense when and how people are occupying the buildings. By leveraging this information, buildings can make smart decisions based on recently developed methods. In this paper, we provide a comprehensive overview of recent advancements in Internet of Things (IoT) technologies that have been designed and used for the monitoring of indoor environmental conditions within buildings. Using these technologies is crucial to gathering data about the indoor environment and determining the number and presence of occupants. Furthermore, this paper critically examines both the strengths and limitations of each technology in predicting occupant behavior. In addition, it explores different methods for processing these data and making future occupancy predictions. Moreover, we highlight some challenges, such as determining the optimal number and location of sensors and radars, and provide a detailed explanation and insights into these challenges. Furthermore, the paper explores possible future directions, including the security of occupants’ data and the promotion of energy-efficient practices such as localizing occupants and monitoring their activities within a building. With respect to other survey works on similar topics, our work aims to both cover recent sensory approaches and review methods used in the literature for estimating occupancy.

## 1. Introduction

The significance of energy efficiency in buildings has become critical in addressing the challenge of climate change and reducing carbon emissions. Buildings are accountable for a substantial portion of global energy consumption and carbon emissions, equating to almost 40% of total energy usage and 36% of total carbon dioxide emissions [1]. Therefore, enhancing the energy efficiency of buildings has become a crucial priority for policymakers, building owners, and operators.

The importance of energy efficiency in buildings can be observed from various perspectives [2,3]. Energy-efficient buildings can diminish energy usage and lower energy bills, which can result in cost savings. Building owners and operators can increase their profitability by reducing energy waste and optimizing building systems. This is especially critical for building owners seeking to attract and retain tenants, as energy-efficient buildings that also improve the usage of building systems provide lower operating costs and more comfortable indoor environments. Energy-efficient buildings can help reduce greenhouse gas emissions and mitigate the effects of climate change. The building sector accounts for a significant amount of carbon emissions, and reducing energy consumption in buildings can help decrease the sector’s impact on the environment. This can be achieved through various measures such as, among others, better insulation; optimized heating, ventilation, and air conditioning (HVAC) systems; and installation of renewable energy systems. Energy-efficient buildings can create a more sustainable and resilient built environment. Energy-efficient buildings, with the smartness they must demonstrate, can help generate more comfortable and healthy indoor environments for occupants by decreasing energy waste and improving building performance. This is especially important in global health crises, such as the COVID-19 pandemic, as energy-efficient buildings can help ensure better indoor air quality and minimize the risk of infectious disease transmission. Energy efficiency in buildings can help create green jobs and support the development of a low-carbon economy. By investing in energy-efficient buildings, building owners and operators can create new job opportunities and contribute to the growth of the green economy. This is especially significant for governments and policymakers, who aim to generate new jobs and promote economic growth while reducing carbon emissions.

Energy efficiency can be achieved in buildings thanks to the Internet of Things (IoT) [4] technologies that have been spreading so much in the last few years. Such technologies enable the creation of so-called smart buildings (SBs) [5] and cognitive buildings [6], which are buildings augmented with sensing/actuating, elaboration, and cognitive capabilities. Among the operations that SBs can perform to greatly assist in reducing energy consumption in buildings, occupancy prediction plays an important role [7,8]. By predicting occupancy, building managers/owners can more effectively manage building systems and reduce energy waste by accurately forecasting whether the building (or a specific room) is occupied and also how many people are occupying a precise area [9]. For instance, with few occupants, building systems like lighting and HVAC can be adjusted or switched off in some parts of the building to save energy. This decreases energy consumption and prolongs the lifespan of building systems and equipment. Moreover, occupancy prediction can optimize the utilization of building systems. By automatically adapting building systems according to the number of occupants, energy consumption can be lowered without affecting comfort levels. For instance, an HVAC system can be calibrated to maintain a pleasant temperature depending on how many people are in the room rather than sustaining a constant temperature. Additionally, occupancy prediction can identify underutilized areas of a building, enabling building operators to optimize space usage and diminish energy consumption. For example, if a particular area of a building is frequently unoccupied, building operators may choose to repurpose that space or decrease the energy consumption of that area. Occupancy prediction can provide valuable information on building usage and assist building operators in making informed decisions about building systems and energy consumption, resulting in substantial reductions in energy waste and expenses.

A wide range of monitoring methods belonging to the IoT, including cameras, sensors, radars, and wearable technology, have been employed in research to monitor room environments, each offering unique capabilities for data collection and monitoring [10]. Once data have been collected from these monitoring techniques, various methods are applied to predict building occupancy. These methods process the collected data and generate occupancy predictions by considering the complex interplay of factors within the building environment. By employing these sophisticated techniques, building managers can optimize energy efficiency, resource allocation, and space utilization based on anticipated occupancy patterns.

Although considerable research has been conducted on occupancy prediction in buildings, in our understanding, there are still gaps in this area that need to be addressed. These gaps may include identifying and resolving limitations or drawbacks of existing methods and exploring the potential benefits of new or emerging technologies for improving accuracy or reducing costs. Furthermore, most of the research has been focused on specific types of buildings or environments, highlighting the need for additional research to evaluate the effectiveness of occupancy prediction methods in a broader range of settings. In addition, there may be opportunities to integrate occupancy prediction with other building automation and control systems, to optimize energy use, comfort, and safety. Therefore, there is still much work to be done in this field, and ongoing research is necessary to advance our knowledge and capabilities in predicting building occupancy.

The aim of this paper is, firstly, to provide an in-depth overview of the recently developed environmental monitoring technologies for indoor environments within buildings that have been ignored in prior literature. Secondly, this paper has the purpose of exploring the various approaches that can be used to extract valuable information for occupancy prediction from these monitoring techniques. Thirdly, it aims to describe some challenges and future directions in the field of occupancy prediction in indoor environments.

The conceptual framework of this review paper is depicted in Figure 1. It comprises several principal axes: (i) the recent data collection technologies involved in monitoring the environment that can sense the presence of occupants or their number; (ii) the most commonly used methods to process these data, namely analytical, machine learning (ML), deep learning (DL), and other methods; (iii) possible challenges that can be found; and (iv) future directions that may emerge.

Figure 2 presents a flowchart that illustrates a typical occupancy prediction process. It begins with data collection methods and progresses to data preprocessing to prepare the data for algorithm implementation. After preprocessing the data, the next stage is to implement different algorithms, and finally, the performance measurement of these algorithms becomes possible to calculate.

The rest of the paper is structured as follows. Section 2 makes a comparison of previous review papers on occupancy detection, estimation, and prediction. Section 3 illustrates the strategy we followed when for searching the work reviewed in this paper. Section 4 gives a comprehensive overview of the various technologies employed for data collection. Section 5 briefly delves into data analysis techniques. An in-depth exploration of challenges and future research directions is presented in Section 6. Finally, Section 7 summarizes the conclusions of the work.

## 2. Previous Review Papers

In sustainable SB design, focusing on reducing energy consumption through understanding occupant behavior, particularly occupancy prediction, has gained prominence. This burgeoning interest is reflected in recent scholarly reviews that underscore the pivotal role of diverse sensor and smart object technologies. Furthermore, these reviews illuminate the critical importance of innovative methodologies, integrating artificial intelligence and analytical strategies to advance the field of energy-efficient building management. Table 1 provides a detailed enumeration of recent review papers that used these smart technologies and methodologies, illustrating their respective roles and functions in the study.

In a recent study, a comprehensive literature review explored the state-of-the-art of people counting and detection in large non-residential buildings [11]. The study critically analyzed various occupancy monitoring methods, outlining their advantages and limitations and making comparisons. However, the study did not extensively explore the significance of recent ultra-wide band (UWB) radar-based occupancy prediction or thoroughly examine ML and DL techniques that play an important role in predicting occupancy in energy-efficient SB environments.

Similarly, another review paper analyzed ventilation conditions and airborne particulate levels in offices by analyzing CO_2_ concentrations for predicting occupancy [12]. While acknowledging the significance of carbon dioxide levels in urban office spaces, the review consistently ignored the importance of recent technology development. It did not provide in-depth information on sensor fusion approaches for building occupancy prediction.

Furthermore, another review paper analyzed six monitoring methods, including different sensors and cameras, to guide selecting appropriate monitoring methods in energy-efficient building environments [13]. The review summarized and discussed the advantages of deterministic schedules, stochastic schedules, and machine-learning methods for improving occupancy prediction accuracy. However, it had limitations, as it did not consider the importance of DL in occupancy estimation and overlooked the significance of edge-based techniques.

Additionally, another review paper provided an overview of various techniques used to predict building occupancy information [14]. The study classified the analyzed studies into three categories: analytical, data-driven, and knowledge-based methods. While extensively covering different sensors and methods for estimating room occupancy, the review failed to address privacy concerns associated with the data.

Other authors conducted a qualitative review of occupancy prediction and detection, comparing and summarizing various solutions based on criteria like performance, cost, and limitations [10]. However, the study did not explore the importance of DL and edge computing techniques for detecting and estimating room occupancy. Furthermore, it did not explain the concept of UWB and other wearable technologies that are highly effective in occupancy estimation and detection in energy-efficient building environments.

The paper in [15] effectively highlighted the role of IoT technology and hybrid ML algorithms in enhancing the accuracy of building occupancy predictions, which is crucial for energy optimization and sustainability. However, it could have benefited from a deeper analysis of the practical challenges and limitations associated with deploying IoT in real-world settings. While comprehensive, exploring data collection methods and predictive algorithms would be enriched by including case studies or real-world applications to demonstrate the practical impact of these advancements.

The review paper in [16] critically examined the shift towards occupant-centric control-based prediction in buildings, highlighting the gap between research advancements and their practical application. It aimed to analyze these controls’ strengths, implementation requirements, and future research directions. While identifying key barriers like computational complexity and data challenges, the study would have benefited from suggesting specific solutions to enhance the real-world adoption of such technologies.

The article in [3] highlighted the significance of occupancy forecasting in buildings to enhance energy efficiency and occupant comfort. The study explored IoT-based monitoring methods for detecting user presence, examining DL and ML algorithms for building occupancy prediction. It aimed to provide a comprehensive overview of current research and future directions in occupancy forecasting, underscoring its potential to save energy, improve security, and ensure safety in building environments. Although the work is effective, the number of articles included in the review is limited.

Despite these limitations, several other reviews have also conducted surveys related to occupancy prediction to maximize energy efficiency, increase comfort levels, and optimize resource use [17,18,19,20,21]. However, these reviews also have deficiencies, such as insufficient detail on censoring techniques, inadequate coverage of recent sensor approaches, and a need for improvement in portraying conventional methods for estimating and detecting room occupancy.

Considering the limitations highlighted above, this paper aimed to investigate various up-to-date sensing techniques; their data collection methods, advantages, and limitations; and when to employ them for data collection to achieve energy-efficient building environments. Moreover, after collecting data from sensors, selecting a suitable approach for detecting and predicting occupancy can also be challenging, due to various factors such as the type of data collected, the required accuracy level, the available computing resources, and the desired outcome. Considering this, the current review delves into various contemporary methods for predicting occupancy, including deterministic and stochastic schedules, ML, and DL algorithms. Furthermore, we elucidate the benefits and drawbacks of these approaches. The decision about which technique to use should be based on assessing the gathered data and each method’s unique advantages. Moreover, this review also investigates challenges and possible research directions for future studies aiming to enhance the energy efficiency of SBs through the use of occupancy predictions.

**Table 1 sensors-24-03276-t001:** Summary of the recent literature review papers.

References	Monitoring Techniques					Mathematical Occupancy Models		ML Occupancy Models			DL Occupancy Models			Transfer Learning	Federated Learning	Challenges
	UWB Technology	Mobility Sensors	Non-Mobility Sensors	Cameras	Electric Meters	Deterministic Schedule Method	Stochastic Schedule Method	SVM	Regression Method	KNN	CNN	RNN	LSTM			
[11], 2021	✕	✕	✓	✓	✕	✕	✕	✕	✕	✕	✕	✕	✕	✕	✕	✓
[12], 2022	✕	✓	✓	✓	✕	✕	✕	✕	✕	✕	✕	✕	✕	✕	✕	✓
[13], 2021	✕	✓	✓	✓	✕	✓	✓	✓	✓	✓	✕	✕	✕	✕	✕	✓
[14], 2020	✕	✓	✓	✓	✓	✓	✓	✓	✓	✓	✓	✓	✓	✓	✕	✓
[10], 2018	✕	✓	✓	✓	✓	✓	✓	✓	✓	✓	✓	✓	✓	✓	✕	✓
[17], 2021	✕	✓	✓	✓	✓	✓	✓	✓	✓	✓	✓	✓	✓	✓	✕	✓
[18], 2020	✕	✓	✓	✓	✓	✕	✕	✓	✓	✓	✓	✓	✓	✓	✕	✓
[19], 2022	✕	✓	✓	✓	✓	✕	✕	✓	✓	✓	✓	✓	✓	✕	✕	✓
[20], 2022	✕	✓	✓	✓	✓	✕	✕	✓	✓	✓	✕	✕	✕	✓	✕	✓
[21], 2022	✕	✕	✕	✕	✓	✕	✓	✕	✕	✓	✕	✕	✕	✕	✕	
[3], 2023	✓	✓	✓	✓	✕	✕	✕	✓	✓	✓	✓	✓	✓	✓	✓	✓
[22], 2023	✕	✓	✓	✓	✕	✕	✕	✓	✓	✓	✓	✓	✕	✕	✕	✓
[15], 2024	✓	✓	✓	✓	✓	✕	✕	✓	✓	✓	✕	✕	✕	✕	✕	✓
[16], 2024	✕	✓	✓	✓	✕	✓	✓	✕	✓	✓	✕	✓	✓	✓	✓	✓

## 3. Search Strategy

In order to find the works for our survey paper, we investigated the following digital libraries: Google Scholar, Scopus, ScienceDirect, Elsevier, IEEE Xplorer, and Springer databases. Our search focused on topics related to occupancy detection, occupancy prediction, occupancy estimation, and energy-efficient buildings. Based on our search strategy, we chose the following keywords: “occupancy detection”, “occupancy prediction”, “occupancy forecasting”, and “Smart Buildings”. Our most recent search for articles was conducted in March 2024. Here are the inclusion criteria that we used to select the publications.
We included articles and book chapters published in English from 2004 to 2024.

Then, we also identified two exclusion criteria that were used to eliminate publications not relevant to this work.
we excluded articles that were based solely on the personal opinions of individual experts;we excluded conference posters, abstracts, short articles, and unpublished works.

Full-text articles were assessed to meet the following eligibility criteria:research articles;articles entirely written in English;occupancy detection;occupancy estimation;occupancy prediction;occupancy sensing techniques/methods;energy-efficient buildings;smart buildings.

We initially looked for articles in the libraries mentioned before based on the identified keywords, and, after applying the search strategy and the eligibility criteria above, we identified and included 97 relevant and valuable articles for this review paper.

## 4. Data Collection Methods

This section comprehensively explains the latest technologies applicable to occupancy prediction in SBs, encompassing UWB radar technology, mobility sensors, non-mobility sensors, sensor combinations, cameras, and smart meters. Furthermore, each subsection of these technologies includes an in-depth examination of their respective advantages and limitations, meticulously presented within dedicated tables for comprehensive analysis. Figure 3 showcases the set of data collection techniques employed to monitor environmental conditions for occupancy prediction within buildings.

### 4.1. UWB Radar Technology

UWB radar is an advanced sensing technology that utilizes short-duration electromagnetic pulses to detect and measure objects within its range [23]. When applied to occupancy prediction, UWB radar becomes a valuable tool for estimating and detecting the presence and movement of occupants in buildings. By emitting UWB pulses and analyzing the resulting reflected signals or echoes, the radar system can gather information about the distance, position, and movement of objects, including people. Examples of UWB radar devices used for occupancy detection, estimation, and prediction are shown in Figure 4. Different data analysis techniques extract relevant data, and occupancy prediction algorithms are employed to determine the occupants’ presence, location, and quantity. UWB radar provides high accuracy and robustness, enabling it to detect occupants even in stationary or partially obstructed conditions. Additionally, UWB radar is non-intrusive and operates at low power, making it well suited for deployment in various building environments. Leveraging UWB radar technology in occupancy prediction systems facilitates optimized building management, enhanced energy efficiency, improved security measures, and a personalized occupant experience. A summary of the recent literature using this technology for occupancy prediction is shown in Table 2.

UWB technology has gained significant attention in predicting occupancy in SBs [24]. The authors of [25] explored the possibility of using cost-effective and dependable technologies for estimating and predicting the number of people in SBs. The research analyzed human body presence patterns through WiFi, UWB, and light signals. It put forth an effective and straightforward approach for utilizing these patterns to count people as they enter through a doorway. This technique can count up to four individuals walking through the door simultaneously, even in densely populated environments. Similarly, in another study [23], the authors presented a technique for counting individuals who pass through a doorway, which involves utilizing impulse radio ultra-wide band (IR-UWB) radars. The proposed system consists of two IR-UWB radars positioned horizontally apart to create a delay effect when an individual walks by. This effect enables the system to detect the direction of movement and subsequently determine the number of individuals in a room. The system’s accuracy was verified by conducting several tests, which demonstrated an accuracy rate of approximately 90%, thereby confirming the system’s effectiveness. Additionally, another study [26] presented a method for detecting and predicting room occupancy using UWB radar based on principal component analysis (PCA). The proposed solution was tested with up to two individuals within the radar’s detection range. The algorithm was able to determine room occupancy with a 100% accuracy rate. Lastly, the research paper in [27] examined the potential of UWB Doppler radars for detecting occupants along with recognizing activities of daily living (ADLs) in smart homes. The study discovered that with a basic setup and conventional feature engineering, a small group of UWB radars could accurately identify ADLs in a practical home environment. The random forest algorithm achieved an accuracy of 80% with an F1-Score of 79% and a Kappa of 77%. These findings demonstrate that radars have promising potential as a research area for smart home applications.

UWB radar technology allows for precise and accurate occupancy predictions within buildings. This technology has the capability to accurately identify and monitor individuals, especially in intricate indoor settings where conventional sensors may encounter difficulties due to barriers or disruptions. UWB radar systems have the ability to accurately record intricate movement patterns and distinguish between numerous sorts of movements, including walking, sitting, and standing [27]. The high level of precision achieved allows for efficient utilization of building spaces, promotes energy economy by optimizing heating and cooling systems according to occupancy, and enhances overall building management and security. However, the higher cost of UWB radar technology compared to other sensing technologies limits its use for occupancy prediction in buildings. Deploying UWB radar systems across a building involves substantial expenditure on hardware, installation, and maintenance. In certain locations with dense barriers or reflective surfaces, UWB radar systems may encounter difficulties in reliably detecting inhabitants due to signal reflections and interference. In addition, the extensive tracking capabilities of UWB radar technology may give rise to privacy problems, necessitating a thorough evaluation of privacy precautions and adherence to rules, such as data protection laws.

**Table 2 sensors-24-03276-t002:** Summary of the papers in which the UWB radar technology was used.

	References	Aim of the Paper	Used Methods	Advantages	Limitations
**UWB** **radar** **technology**	[23]	Counting people	N/A	High precision, non-intrusive, multi-person detection, real-time monitoring, robust to environmental factors, low power consumption, easy installation, versatile option for occupancy	Limited range, expensive, complex to understand, signals can be interrupted
	[25]	Estimating and predicting people in SBs	hierarchical algorithm		
	[26]	Detecting and predicting room occupancy	PCA based algorithm		
	[27]	Activity recognition	KNN, Random Forest, Classification and Regression Tree		

### 4.2. CO_2_ Sensor

The concentration of carbon dioxide in the air can be measured by a CO_2_ sensor, which can also be utilized to detect room occupancy. As people exhale CO_2_, the concentration of carbon dioxide in a room increases with the number of individuals present. Consequently, by analyzing the CO_2_ levels in the air, a CO_2_ sensor can be employed to estimate occupants and their numbers in a building/room. Examples of CO_2_ sensor devices used for occupancy detection, estimation, and prediction are shown in Figure 5. A summary of the recent literature using CO_2_ sensors is shown in Table 3.

Several papers explored the use of CO_2_ concentrations to estimate room occupancy in SBs. In the research paper in [28], the authors explored methods for predicting room occupancy using a dynamic neural network model that relied on carbon dioxide levels in a room where the number of occupants varied irregularly. The model was trained and tested using a time-delay neural network and found to have a high level of accuracy. Another study [29] developed a systematic approach for managing the HVAC system in a classroom on a university campus, which relied on monitoring the concentration of CO_2_ in the surrounding air. To accurately predict CO_2_ levels, the researchers evaluated six state-of-the-art ML algorithms and customized them for this specific purpose. Their multilayered perceptron network demonstrated superior performance among the tested algorithms, due to its remarkable capacity for learning nonlinear relationships in the CO_2_ data. A simple method for detecting occupant numbers was proposed by Szczurek et al. using statistical pattern matching [30]. Their analysis was based on CO_2_ concentration statistics, such as a correlation coefficient and autocorrelation function, to determine occupancy levels in a half-hour window. Using this method, one can estimate mean occupancy levels over a long period, e.g., half an hour, which is a long time in real-time system control. Explicit modeling of CO_2_ concentrations based on occupant numbers is difficult and inaccurate, due to the complex nature of indoor CO_2_ concentrations. One can directly model the relationship between inputs and outputs using data-driven approaches like ML algorithms. In their study, Zuraimi et al. [31] compared dynamic physical models with ML methods such as support vector machines (SVMs), prediction error minimizations, and artificial neural networks (ANN) for estimating building occupancy based on the concentration of CO_2_. Experiments in a room with 200 people showed that the ANN and SVM approach performed best. Rahman and Han, in [32], investigated neural networks and compared them to a Markov chain Monte Carlo (MCMC) algorithm for estimating occupancy using CO_2_ levels. Under certain conditions, both models produced satisfactory results. Based on data collected from a different room, Ebadat et al. developed a gray-box model to simulate CO_2_ dynamics in a new room [33]. The regularized deconvolution model in [34] was employed to estimate a new room’s occupancy in the building. According to Alam et al. in [35], ANNs are a useful tool for occupancy estimation using CO_2_ data. Various occupancy profiles and airflow schemes were simulated to optimize their occupancy estimation system. The results of this study can be used as guidelines for estimating occupancy using ANNs with CO_2_ concentrations. In the study in [36], the CNN-XGBoost DL method was proposed to predict occupancy based on indoor climate data, and its performance was compared with supervised and unsupervised ML algorithms and artificial neural networks.

An advantage of utilizing CO_2_ sensors for occupancy prediction in buildings is that it not only helps to give information about whether the room or buildings are occupied or not, but it also gives information about how many people are inside a room [8]. As people live in enclosed areas, they release CO_2_ through respiration, causing an increase in atmospheric CO_2_ levels. Sensors that measure CO_2_ levels can detect people’s presence in a room or building. This technique is especially valuable in environments where conventional motion detectors may not be feasible or efficient, such as spaces with restricted activity or where people may stay still for long periods, such as meeting rooms, educational facilities, or recreational areas. Additionally, CO_2_ sensors are very affordable and straightforward to install, which makes them a cost-efficient option for detecting and managing occupancy in buildings. CO_2_ sensors used for occupancy prediction in buildings have a drawback, in that they cannot offer real-time monitoring or that the response to the number of people is very slow. Furthermore, other factors, such as high ventilation rates or widespread CO_2_ emissions from sources unrelated to occupancy, may greatly influence CO_2_ levels, making CO_2_ sensors inappropriate for certain locations. In addition, CO_2_ sensors necessitate frequent calibration and maintenance to guarantee precise measurements, and their efficiency may decline over time if not adequately upheld. Therefore, CO_2_ sensors should be employed in conjunction with supplementary sensors or technologies to achieve comprehensive occupancy prediction and management in buildings.

**Table 3 sensors-24-03276-t003:** Summary of the papers in which CO_2_ sensors are used.

	References	Aim of the Paper	Used Methods	Advantages	Limitations
**CO_2_** **sensor**	[28]	Predicting room occupancy	dynamic neural network	Not-intrusive, cost effective, ability to estimate many occupants, longer lifespan, direct relationship with occupancy, can improve indoor air quality by favoring ventilation,	Time delay, can produce false positive if there is something else in room (e.g., pets)
	[29]	Managing the HVAC system in classroom	ML algorithms		
	[30]	Detecting occupants number	statistical pattern matching		
	[31]	Estimating building occupancy	SVM, ANN		
	[32]	Estimating occupancy	MCMC, and neural networks		
	[33]	Multi room occupancy estimating	grey box model		
	[35]	Occupancy estimating	ANN		
	[36]	Occupancy predicting	CNN-XGBoost		

### 4.3. Passive Infrared Sensor

A passive infrared sensor (PIR) is a sensor that can identify human body heat within its field of view by detecting changes in infrared radiation emitted by objects within its range. Example PIR sensors used for occupancy detection, estimation, and prediction are shown in Figure 6. SBs can use PIR sensors to estimate room occupancy [37]. Placing PIR sensors in different locations within a room or a building allows detecting changes in the motion of people present or when people enter or leave an area. A central control system receives sensor data to infer occupancy patterns. To increase accuracy and reduce false positives, this system can be combined with other types of sensors, such as CO_2_ or light sensors.

PIR sensors are gaining popularity for estimating room occupancy in SB environments. A list of recent papers that used PIR sensors is given in Table 4. An occupancy detection system using PIR sensors was proposed by Dodier et al. in [38]. They deployed three PIR sensors in their system to detect occupant presence. After that, Bayesian probability theory was applied to determine whether a zone has any occupants. Duarte et al. used PIR sensors to monitor long-term changes in occupancy levels between different rooms [39]. As a result, the authors analyzed the occupancy patterns detected by the detector and compared them with standard occupancy diversities as specified in [40]. This resulted in significant differences between the real occupancy patterns and the standard occupancy diversities commonly used in energy simulation tools. In the article in [41], Liu et al. presented a sensor-based occupancy detection system. Occupancy was detected using a hidden Markov model (HMM) with expectation maximization.

Various studies have used PIR sensor activity directly as evidence of occupant presence in building occupancy models [42,43]. In one study, Wahl et al. developed a PIR-based occupancy counting system that tracks the movement directions of occupants [44]. To estimate occupancy with the movement directions of occupants, two simple methods based on directions and a probabilistic method based on distance were proposed. Using a PIR sensor, researchers estimated the number of people in a room [45]. In a first step, they used an infinite HMM model to extract the motion patterns of occupants from raw sensory data. Statistical regression models were used to estimate occupancy based on the extracted patterns. The authors of the study in [46] proposed a new method that utilizes a single PIR sensor for detecting human presence, which yielded promising results. The study suggests this approach could be used to monitor and track at-risk patients in indoor settings.

**Table 4 sensors-24-03276-t004:** Summary of the papers in which PIR sensors were used.

	References	Aim of the Paper	Used Methods	Advantages	Limitations
**PIR** **sensor**	[38]	Occupancy detection	Bayesian probability theory	Not-intrusive, low cost, high accuracy, real time occupancy detection, can be easily installed	Lack of identification, not suitable for static object, limited coverage area
	[39]	Monitor long term changes in occupancy	Stochastic model		
	[41]	Occupancy detection	HMM		
	[44]	Tracks occupants’ movement directions	probabilistic method		
	[45]	Estimate number of people	HMM and Statistical regression models		
	[37]	Estimate room occupancy	regression controller		

An important advantage of PIR sensors is their capacity to identify the presence of humans by analyzing body heat and movement [46]. Warm objects, like humans, emit variations in infrared radiation that PIR sensors detect as they move within their detection range. PIR sensors are very efficient in detecting occupancy in indoor areas, due to their ability to rapidly and reliably detect movement without requiring physical contact with inhabitants. PIR sensors are cost-effective, energy-efficient, and simple to install, which contributes to their widespread use in occupancy detection systems across different building settings. Due to their limited detection range and broad field of view, PIR sensors have some limitations in forecasting occupancy in buildings. PIR sensors generally have a limited detection angle and range, which can result in blind spots or undetected occupancy. Furthermore, environmental factors such as temperature fluctuations, ventilation, or the presence of pets or other moving objects can easily trigger PIR sensors, resulting in false positives. This can result in imprecise occupancy forecasts and potentially wasteful utilization of building resources if systems are activated needlessly. In addition, PIR sensors may not be appropriate for detecting individuals who are not moving or engaging in modest activities, such as typing or reading, which can impact the precision of occupancy forecasts in specific situations. To improve the accuracy and reliability of occupancy prediction, it is advisable to combine PIR sensors with other sensing technologies, even though they are useful for detecting occupancy in buildings.

### 4.4. Sensor Combination

Sensor combination, also called sensor fusion, is a technique used to improve the overall performance and dependability of detecting occupancy in SBs. It can combine data from different types of sensors like PIR, CO_2_, light, and sound sensors to gather comprehensive information about a room or building, allowing for better detection of occupancy patterns [47]. Sensor fusion combines the data collected from each sensor into a unified system that applies algorithms and ML methods to analyze the data and identify patterns, enabling the system to make more accurate decisions about room occupancy. A summary of the recent literature that has used sensor fusion is shown in Table 5.

The study in [48] utilized different machine-learning algorithms to examine how combining data from multiple sensors can enhance the quality of the information. It is widely recognized that accurate occupancy estimation can be valuable in a variety of situations and applications. An occupancy detection system based on an environmental sensor system composed of CO_2_, light, temperature, and humidity was used in the approach proposed by Candanedo and Feldheim [49]. Detection was achieved using three algorithms: linear discriminant analysis (LDA), classification and regression trees (CART), and random forest (RF). The researchers concluded that satisfactory occupancy detection results can be achieved with the right selection of features and algorithms. Based on the assumption that the ground truth occupancy is unavailable, Candanedo et al. evaluated an HMM occupancy algorithm using the same sensors [50]. In this study, different combinations of features were used to test the HMM. They reported that the best detection performance came from HMMs with CO_2_ features. Ertugrul proposed a recurrent extreme learning machine (ELM) approach that utilized the same sensors as in [49] to detect occupancy based on the temporal dynamics of environmental time series data. In [51], Kraipeerapun and Amornsamankul developed modified stacking schemes with a dual output neural network to determine occupancy using environmental sensors measuring CO_2_, light, temperature, and humidity. Two stages are involved in the proposed approach. The first stage consists of the training of multiple neural networks and the concatenation of their outputs into the second stage, which is called the concatenation stage. In the second stage, two different occupancy detection techniques were discussed.

A new occupancy estimation system was presented by Masood et al., which utilized environmental parameters, such as CO_2_, temperature, humidity, and pressure, using the ELM algorithm [52]. A feature selection algorithm based on ELM was proposed because of ELM’s extremely fast learning speed. Experimental results showed that their proposed ELM-based wrapper method outperformed popular filter methods. As a result, they concluded that pressure sensors, which are rarely used for occupancy estimation, are meaningful. Another study by Masood et al., which incorporated a filter-wrapper hybrid feature selection method with ELM, estimated the range of occupants as zero, low, medium, and high [32]. First, they ranked all features using a relative information gain approach and then used an incremental search to select the best features to estimate occupancy. The authors also presented a hybrid ELM with a hybrid feature-scaled layer that contained a dynamic feature extraction layer to estimate occupancy using the same environment sensors as in [53]. In a conventional static feature selection process, static features were selected using a filter. After selecting static features, the dynamic features were combined with the static features to form a new set of features. As a result, they could guarantee a high accuracy, as well as a fast speed. An occupancy estimation system based on CO_2_, temperature, and humidity data was presented by Szczurek et al. in [54]. A wrapper-based method was used for feature selection; and for estimation, two K-Neareast Neighbor (KNN) and LDA learning algorithms were used. As part of their experiments, they investigated the estimation of occupancy based on individual sensor data and the fusion of multiple sensor data. Based on the authors’ findings, the KNN-trained classifier was much more effective than the LDA-trained classifier. The authors propose a two-stage ELM method for occupancy estimation using environmental sensors of CO_2_ and temperature [55]. For the first stage, local representations of raw features were obtained using the ELM algorithm. Afterward, a linear SVM was applied to map local representations to occupant numbers. The study in [48] involved using various sensors in different rooms with different occupancy levels, and ML models were employed to analyze the resulting data. The researchers carefully considered privacy concerns and the Hawthorne effect to ensure the data collected were realistic. The study demonstrated the importance of utilizing different types of sensors and multiple rooms when gathering data. The study’s results suggest that incorporating a variety of sensors can enhance the accuracy of machine-learning models. Another paper [56] proposed a technique for using data fusion to detect whether individuals are present or absent in a room. The technique involves collecting data from a range of sensors, such as temperature and humidity sensors, to identify the presence of a person. Using the principles of evidence theory, the data collected from each sensor can be considered as evidence. This approach provides a comprehensive method for detecting the presence of individuals in a room, showing high accuracy. Additionally, the objective of another study [57] was to achieve practicality and cost-effectiveness in deploying a system by utilizing a smart sensing network, embedded passive infrared (PIR)/CO_2_ sensors, and environmental system modeling. The approach aimed to support sustainable building operations, while ensuring comfortable thermal conditions and acceptable indoor air quality. The system integration was designed to be simple and concise, making it easier and more efficient to implement. Using these technologies and models, the study sought to promote environmentally conscious building operations, while maintaining occupant comfort and health.

By combining multiple sensors, occupancy prediction systems can achieve higher accuracy [8]. Each type of sensor has unique advantages and disadvantages, and their combination can compensate for each individual sensor’s limitations. PIR sensors excel at detecting motion but may fail to identify stationary occupants, whereas CO_2_ sensors offer indirect indications of occupancy levels. By integrating these sensors, it becomes possible to enhance the accuracy and dependability of occupancy predictions through the comparison and analysis of data obtained from various sources. Combining sensors ensures redundancy in detecting occupancy. In the event of a sensor failure or false data production caused by environmental conditions or technological problems, the remaining sensors within the system can still contribute to the estimation of occupancy. This duplication improves the occupancy forecasting system’s dependability and reduces the likelihood of incorrect alerts or missed detections. The distinct qualities and problems present in various building contexts can influence occupancy prediction. Occupancy prediction systems can enhance performance in different places, such as large offices, conference rooms, restrooms, and hallways, by utilizing several sensors to adapt to varying conditions. The versatility of the system guarantees that occupancy prediction remains successful in varied circumstances, hence maximizing its utility. Incorporating several sensors into an integrated occupancy prediction system increases the complexity of the system design, installation, calibration, and maintenance. Each type of sensor incurs specific costs for its hardware, installation, and maintenance, which collectively contribute to the total cost of the system. Efficiently combining many sensor types into a single occupancy prediction system necessitates meticulous strategizing and synchronization. Ensuring compatibility among numerous sensors, establishing uniform data formats, and coordinating sensor readings can be difficult, particularly when working with diverse sensor networks from several manufacturers or with distinct communication protocols. These integration hurdles may impede the smooth functioning of the occupancy prediction system and necessitate continuous maintenance and troubleshooting endeavors to resolve compatibility issues or data anomalies.

**Table 5 sensors-24-03276-t005:** Summary of the papers in which sensor fusion was used.

	References	Aim of the Paper	Used Methods	Advantages	Limitations
**Sensor Fusion**	[47]	Occupancy prediction using DL	DL networks	Non-intrusive, enhanced accuracy, helpful for energy management, helpful in security monitoring, increased robustness	Complexity of integrating, increase overall cost, data analysis time increase
	[48]	How occupancy estimation can benefit from sensor fusion	ML models		
	[57]	Proposed smart sensing network for cost-effectiveness	ML models		
	[56]	Occupants detection in a room	evidence theory		
	[54]	Occupancy estimation	KNN		
	[49]	Occupancy detection and prediction	Linear discriminant analysis, classification and regression tree, random forest		
	[50]	Occupancy detection and prediction	HMM		
	[52]	Occupancy estimation	ELM		
	[55]	Proposed ELM method for occupancy estimation	ELM		

### 4.5. Non-Mobility Sensors

Non-mobility sensors refer to sensors that can detect the presence of people in a room without relying on their movement. These sensors detect changes in the surrounding environment when someone enters or exits a room. Non-mobility sensors can be utilized in various ways to estimate the occupancy of a room, and some examples of their applications are listed below.

#### 4.5.1. Bluetooth Low Energy

Bluetooth low energy (BLE) is a technology that can estimate room occupancy by sensing the presence of Bluetooth-enabled devices, such as smartphones or wearables, in a designated area. BLE devices emit a signal called a beacon, which can be picked up by BLE receivers placed in the room. As soon as a BLE device comes into range of a BLE receiver, the receiver detects the beacon signal and sends a message to a server or controller that analyzes the data to determine if the device owner is in the room. Examples of BLE devices used for occupancy detection, estimation, and prediction are shown in Figure 7. These data can estimate the number of people occupying the room. A summary of the papers that used this technology is shown in Table 6.

In particular, the authors of [58] proposed a novel methodology that employs BLE communication technology to detect and count room occupancy using pattern recognition. Multiple regression and classification algorithms were employed, and they yielded encouraging outcomes in various settings. The best classifier could accurately detect room occupancy with an average accuracy of 97.97% across all datasets. Similarly, another study [59] proposed a multi-feature KNN classification algorithm that utilizes affordable BLE networks to extract occupancy distribution. To validate the proposed methods, an on-site experiment was carried out in a standard office of an institutional building, which showed very good results. Furthermore, the authors put forth a technique for detecting the presence of individuals using low-impact sensors, including BLE sensors, to estimate room occupancy [60]. The authors proposed an occupancy estimation system based on iBeacon and BLE technology in [61]. Using the iBeacon protocol as a starting point, they modified it to make occupancy detection more accurate and efficient. Based on received signal strength indicators (RSSIs) from different iBeacons, two algorithms, KNN and decision tree (DT), were employed to categorize occupants into different rooms. In [62], iBeacons were used to improve the performance of occupancy estimation. Feature stability for classification was obtained by using a smoothing algorithm. After that, the occupants were classified into different rooms using an SVM algorithm. Similarly, Filippoupolitis et al. applied an SVM algorithm for occupancy estimation using Bluetooth beacons [41]. The authors used statistical features to classify occupancy, instead of raw RSSI values. They also explored the same features in different works and three learning algorithms, namely SVM, KNN, and logistic regression, for occupancy estimation using Bluetooth beacons [63].

**Table 6 sensors-24-03276-t006:** Summary of the papers in which BLE technology was used.

	References	Aim of the Paper	Used Methods	Advantages	Limitations
**BLE**	[58]	Detect and count occupancy	Regression and classification algorithms	Non-intrusive, cost-effective as already in phone, produce real-time data, easy to implement	Limited range, may produce false result if occupants can carry two or more, signal can be interrupted by other signals
	[59]	Proposes a multi-feature classification algorithms	KNN classification model		
	[60]	Occupancy detection	ML, optimization, and probabilistic approach		
	[61]	Occupancy estimation and prediction	KNN and DT		
	[62]	Improve the performance of estimation	Smoothing algorithms		
	[41]	Occupancy estimation	SVM ML model		
	[63]	Occupancy estimation and prediction	SVM, KNN, and regression		

A significant advantage of BLE is its extensive adoption and compatibility with many devices such as smartphones, wearables, and IoT sensors [58]. BLE-enabled devices have the capability to function as proximity beacons, enabling accurate monitoring of individuals as they navigate within a facility. This enables real-time continuous occupancy monitoring and provides valuable insights into space use patterns. Occupancy-prediction systems based on BLE are both cost-effective and straightforward to implement. These systems make use of the existing infrastructure and do not necessitate substantial hardware expenditure. Moreover, battery-powered devices benefit greatly from BLE technology’s low power consumption. This ensures that occupancy detection applications can work reliably for an extended period of time. While BLE technology is capable of accurately detecting the presence of occupants who have BLE-enabled devices, such as smartphones or wearables, it may fail to detect those who do not have such devices or opt to disable Bluetooth capabilities. This constraint may result in insufficient occupancy data and mistakes in prediction systems, particularly in settings with diverse user demographics or certain privacy considerations. Moreover, BLE-based devices may encounter difficulties in precisely discerning between several inhabitants or monitoring individuals in congested areas where many signals overlap. Furthermore, construction materials, furniture, or other electronic devices can cause interference or weakening of BLE signals. This can impact the reliability and precision of occupancy prediction. Thus, although BLE technology provides numerous advantages for occupancy detection, its efficacy can fluctuate based on the particular application and environmental conditions.

#### 4.5.2. Wireless Fidelity (WiFi) Technology

WiFi is a wireless technology that enables high-speed internet and networking between devices using radio waves. To achieve this, WiFi networks use a wireless access point to send data through the air, without requiring physical cables to connect devices. Examples of WiFi devices used for occupancy detection, estimation, and prediction are shown in Figure 8.

The capability for WiFi sensing or WiFi-based occupancy detection enables the estimation of room occupancy within buildings. This process involves utilizing the radio waves that WiFi access points emit to detect the presence of people within a room or building. As people move in a room, their movement causes minor changes in the radio waves, which the access points can detect. By analyzing these changes, algorithms can determine if any individuals are in the room and even estimate the number of people and their locations. A summary of the papers that used this technology is shown in Table 7.

The estimation of building occupancy with the help of WiFi techniques is gaining popularity. The authors of [64] introduced and assessed ML-based techniques for estimating classroom occupancy using data from a dense wireless network on a large university campus. The authors also compared their WiFi sensing approach to that of dedicated beam counters, achieving a satisfactory level of accuracy. Another study [65] investigated the viability and possibility of a crowd estimation system that can anticipate both the quantity and positioning of a crowd. This system utilized a combination of WiFi IoT technology and ML. The researchers conducted practical experiments to evaluate the system, involving up to ten individuals to count the crowd and two actual environments, each divided into four sections for localization over three days. The results of the experiments were very satisfactory. Several studies [32,66,67,68,69] investigated occupancy measurement using WiFi technology. According to [66], the authors designed an end-to-end system that infers occupancy without any intrusions. They determined occupancy and MAC addresses by analyzing packets sent by access points (APs). Taking advantage of the temporal correlation of historical data, the system measures indoor snapshot occupancy in real time on the front end but incorporates historical data on the back end. Due to temporal correlation, their algorithm has a long memory on devices. Errors will increase if the occupant does not bring her mobile device into the building. A recurrent neural network (RNN) was presented in the study in [32] to predict occupancy information stochastically using feedback RNNs. Signals are captured through scanning WiFi connections between WiFi APs and devices owned by occupants. A WiFi-based noninvasive occupancy sensing system was proposed to obtain occupancy data from COTS access points and mobile devices using WiFi traffic analysis in [68]. It achieved 98.85% accuracy in detecting occupancy in a practical office area.

**Table 7 sensors-24-03276-t007:** Summary of the papers in which WiFi technology was used.

	References	Aim of the Paper	Used Methods	Advantages	Limitations
**WiFi** **Technology**	[65]	Occupancy estimation and prediction	ML	Non-intrusive, cost-effective, easy to implement, real-time monitoring	Limited range, may produce false result, signal can be interrupted by other signals
	[64]	Estimating classroom occupancy	ML		
	[32]	Occupancy prediction	RNN models		
	[66]	Occupancy detection	N/A		
	[68]	Proposed occupancy sensing system	DL		

WiFi technology provides numerous benefits for predicting occupancy in buildings [70]. An important benefit is the extensive accessibility and pre-existing infrastructure in the majority of today’s buildings. WiFi connection points are already equipped in indoor spaces, ensuring extensive coverage throughout the entire structure. Occupancy prediction systems utilize WiFi signals to detect and monitor the presence of individuals by leveraging their smartphones, tablets, or other WiFi-enabled devices. This method obviates the necessity for deploying supplementary hardware, thereby diminishing implementation expenses and streamlining installation. Furthermore, occupancy prediction systems that rely on WiFi can utilize sophisticated signal processing methods to analyze the strength of WiFi signals. This enables accurate determination of the exact location of individuals inside the building. This allows precise and immediate monitoring of occupancy, offering a vital understanding of how space is being used. This helps building managers efficiently allocate resources and enhances energy conservation. A drawback of using WiFi for occupancy prediction in buildings is its dependence on user devices and the privacy considerations that come with this. WiFi-based systems utilize signal monitoring of smartphones or other WiFi-enabled devices to track inhabitants, which may have privacy problems. Issues about data collection, monitoring, and consent may occur, especially in situations where individuals are unaware or have not specifically agreed to be observed. Furthermore, WiFi-based occupancy prediction systems may face difficulties in precisely detecting occupancy levels in regions with sparse device concentrations or when individuals do not possess WiFi-enabled devices. This constraint might result in insufficient occupancy data and mistakes in forecasting algorithms, especially in settings with varied user demographics or privacy-conscious residents. Moreover, construction materials, furniture, or other electronic devices can disrupt or weaken WiFi signals, thereby affecting the dependability and precision of occupancy forecasting. Hence, although WiFi technology has numerous advantages for occupancy detection, its efficacy can fluctuate depending on the particular use case, environmental variables, and privacy concerns.

#### 4.5.3. Sound Sensor

A sound sensor is a non-mobility sensor that detects sound waves and translates them into electrical signals. It uses a microphone or comparable device to capture sound waves in the surrounding environment. A sound sensor can be employed to measure the sound level in a space for a certain duration to estimate the occupancy of a room. When individuals are present in the room, they produce sounds through activities such as talking, moving, or interacting with objects. These activities raise the overall sound level in the room, which the sound sensor can detect. Examples of sound sensors used for occupancy detection, estimation, and prediction are shown in Figure 9. By studying the sound level over time, algorithms can be applied to identify patterns and decide whether the room is occupied. A summary of the publications in this field is highlighted in Table 8.

Different authors have produced impressive publications using this type of sensor. The authors in [71] examined the potential applications of a standard electrodynamics loudspeaker. These include real-time monitoring of room occupancy to detect the presence of individuals in the workplace, identification of perimeter changes (such as the opening of a door or window) to alert security of potential intruders, and tracking of thermal anomalies to signal unusual temperature variations. Similarly, the paper in [72] developed an effective method for determining and predicting the number of people in a room. To achieve this, the researchers utilized various low-cost sensors, including CO_2_, temperature, illumination, sound, motion, and ML models. In addition, they utilized PCA to assess the effectiveness of a dataset with fewer dimensions. The ultimate goal was to accurately estimate room occupancy using multiple types of sensors and data analysis techniques. A wireless-sensor meeting room management system, iSense, measured decibel levels using capacitors (microphones) [73]. As a result, iSense could detect the status of a conference/meeting. In addition, sound sensors can be linked with other sensors like ambient sensors to estimate occupancy in buildings [74].

**Table 8 sensors-24-03276-t008:** Summary of the papers in which sound sensor technology was used.

	References	Aim of the Paper	Used Methods	Advantages	Limitations
**Sound** **Sensor**	[72]	Occupancy prediction	ML algorithm	Non-intrusive, cost-effective, no need physical contact, real- time monitoring	Limited range, noise can reduce the quality, contain privacy issues
	[71]	Occupancy detection	Electrodynamic loudspeaker		
	[74]	Occupancy estimation	Sound technology		
	[73]	Occupancy detection	iSense Technology		

An important advantage of sound sensors is their capacity to identify occupancy using audio signals, such as human voices or noise caused by movements [75]. Sound sensors are capable of detecting the presence of individuals in environments where other sensing technologies, like motion sensors, would not work well or be suitable, such as in places with low activity. Sound sensors are well-suited for predicting occupancy in locations such as libraries, study rooms, or private offices. Moreover, sound sensors are very affordable and simple to set up, making them a cost-efficient option for occupancy detection systems. Sound sensors can offer real-time occupancy monitoring and provide valuable information on space utilization by analyzing sound patterns and intensity levels. This allows building managers to optimize resource allocation and improve user comfort and safety. A basic limitation of sound sensors for occupancy estimation in buildings is their vulnerability to external noise and interference. Sound sensors may unintentionally identify sounds that are not linked to occupancy, such as HVAC systems, outdoor noise, or industrial operations. This can result in false positives or inaccurate forecasts of occupancy. This can lead to inconsistent occupancy data and may necessitate the use of supplementary signal processing or filtering methods to differentiate between pertinent and extraneous sounds. In addition, sound sensors may encounter difficulties in detecting occupancy in locations characterized by high levels of background noise or acoustically complex surroundings, such as open-plan offices or spaces with surfaces that cause sound to bounce back and echo. In addition, the use of sound sensors may give rise to privacy problems, since inhabitants may feel uneasy about the possibility of their conversations or activities being monitored. Environmental constraints, interference issues, and privacy concerns can constrain the efficacy of sound sensors, despite their distinct benefits for occupancy prediction. Thus, meticulous implementation and calibration are necessary to guarantee dependable performance.

#### 4.5.4. Camera—Optical Sensor

Optical sensors, typically called cameras, can also be used to detect and monitor human presence in a particular area. Examples of camera devices used for occupancy detection, estimation, and prediction are shown in Figure 10. This method employs computer vision algorithms to analyze video footage and recognize human shapes in the captured frames. By tracking changes in the quantity and position of people within the space, the system can estimate the number of individuals present at any given moment. Camera-based occupancy estimation is known for its high level of accuracy and can also supply supplementary data, such as the whereabouts and motion patterns of individuals in a room. A summary of the papers that used the camera is highlighted in Table 9.

The study in [76] introduced a vision-based technique that employed DL-based algorithms for head detection to estimate the number of individuals in sizable indoor spaces, utilizing multiple cameras. The approach was evaluated in a classroom setting with numerous obstructions, and it demonstrated a remarkable capacity for predicting the number of individuals in the room compared to actual measurements. Similarly, the authors put forward a new indoor occupancy estimation approach consisting of a three-level fusion framework. The study used the scene-knowledge fusion counting method to determine the number of individuals in a room [77]. Additionally, a line-based fusion counting method employed two cameras at the entrance and inside a room to detect the movement of individuals and determine the number of occupants based on the number of passing events. Using a multi-camera system, Fleuret et al. presented a method to calculate the number of indoor occupants based on the locations of all individuals [78]. Liu et al. presented a vision-based occupancy estimation system that uses cameras at entrances and in rooms to estimate occupancy. In addition to static vision algorithms, motion-based algorithms were employed to detect indoor occupancy. Using a dynamic Bayesian network approach, a final occupancy estimation was calculated by combining the estimation results of the entrance and the room. A more advanced version of that work is available in [79]. The authors proposed building occupancy estimation with cameras as a cascade framework. A pre-classifier was used to filter out non-head regions as a first step. Next, the head windows were classified using a convolution neural network (CNN). In the final step, a clustering algorithm fused consecutive frames to detect heads more effectively. The proposed approach achieved a high estimation accuracy of 95.3%. With a camera at the entrance of a room, Petersen et al. developed an occupancy estimation system that counted occupants entering and leaving a room [80]. Using cameras at each zone portal, Tomastik et al. could determine how occupants moved between zones of a building [81]. Additionally, they developed a non-linear stochastic state-space model for estimating occupancy during emergency egress based on occupant movements.

A significant advantage of cameras is their capacity to offer visual verification of occupancy, enabling accurate identification and monitoring of individuals within a given area [82]. Cameras can effectively ascertain the existence and movements of individuals by recording pictures or video footage, enabling real-time monitoring of occupancy and offering valuable information on patterns of space utilization. Computer vision algorithms can be used to analyze this visual data, enabling the detection and counting of occupants, differentiation between various activities or postures (such as sitting, standing, or walking), and even the identification of specific individuals if required. Moreover, the integration of cameras with other sensing technologies enhances the accuracy and reliability of occupancy prediction systems, providing additional data for comprehensive occupancy monitoring. Privacy concerns and ethical reasons are significant constraints on using cameras for occupancy prediction in buildings. Cameras capture visual data, including photos of individuals and their actions, potentially triggering privacy concerns and objections from the subjects. Privacy concerns regarding the collection, retention, and use of personal information may arise, leading to opposition or legislative limitations on camera-based occupancy prediction systems. Moreover, the existence of surveillance cameras might influence the behavior and sense of privacy among individuals, thereby affecting their level of comfort and productivity in the area [11,18,83]. In addition, camera-based systems require sufficient illumination and unobstructed perspectives to operate optimally, restricting their suitability in low-light settings or regions with limited visibility. Furthermore, camera systems might incur substantial charges for installation and maintenance, encompassing costs for hardware, data storage, and system integration. They may face difficulties in reliably detecting occupancy in places with intricate layouts, obstructions, or frequent variations in lighting conditions, which can affect the dependability and precision of forecasts. Hence, although cameras provide potent capacities for forecasting occupancy, their integration must take into account privacy considerations, economic ramifications, and technical constraints, to guarantee ethical and efficient utilization in building settings.

**Table 9 sensors-24-03276-t009:** Summary of the papers in which cameras were used.

	References	Aim of the Paper	Used Methods	Advantages	Limitations
**Camera**	[78]	Calculate number of occupants	Stochastic model	Offer high accuracy, can count occupants, real-time monitoring	Expensive, privacy issues, not easy to install
	[77]	Occupants estimation	Dynamic Bayesian Fusion mechanism		
	[76]	Occupants detection and estimation	DL architectures		
	[84]	Occupancy measurement	Dynamic Bayesian network-based method		
	[80]	Occupancy estimation	Hierarchical algorithm		
	[79]	Occupancy estimation and prediction	CNN		
	[81]	Monitoring occupants movement	Non-linear stochastic state-space model		

#### 4.5.5. Electric Meter

Electric meter-based room occupancy estimation refers to the use of changes in electricity consumption to estimate and predict occupants in a building. This approach operates on the assumption that people use electrical appliances while present in a room, and their activity results in changes in power consumption patterns. Examples of electric meter technologies used for occupancy detection, estimation, and prediction are shown in Figure 11. The electric meter data are collected and analyzed using algorithms that differentiate between normal electricity consumption patterns and those associated with human presence. A summary of the papers that used this technology for data collection is shown in Table 10.

According to the study found in [85], there is a noticeable correlation between occupancy and the amount of electricity consumed in various areas of a building, including office space, corridors, and meeting rooms. The authors of [86] proposed a system to predict room occupancy using only data from smart meters to predict whether a room is occupied. Datasets included appliance state, appliance energy, and house-level occupancy. To determine the accuracy of occupancy detection, they used the F-measure. using Adaboost and random forest classifiers, and a 90% accuracy was achieved. Using off-the-shelf electricity meters, 35 features of the electric load curve were extracted for the study in [87]. The researchers found that PCA and SVMs were the most accurate ML methods, at between 83 and 94%. Over 18 months, the authors of [88] studied more than 5000 households to determine if they were occupied. Several ML techniques were used to achieve an accuracy of 90.1% in predicting current and future occupancy information. The authors of [89] proposed a novel concept of a standby state based on occupant presence or absence to reduce energy consumption. A KNN method was used in the classification process, with an accuracy of 94% based on the F-measure. Using mutual information, researchers reduced the sparsity of datasets in their study [90]. An overall 83.37% occupancy detection F-measure was achieved using random forest classifiers, and an 82.79% occupancy detection F-measure was achieved using DT classifiers. To enhance precision, we need a more complete picture of the weather, local events, and neighborhood [88].

Electric meters can indirectly estimate the occupancy levels within a building by observing electricity usage trends [87]. Occupancy often corresponds to energy usage, since inhabitants make use of electrical devices, lighting, and HVAC systems. Electric meters can offer valuable insights into occupancy trends and patterns by analyzing fluctuations in electricity demand. This information allows building managers to optimize resource allocation, improve energy efficiency, and identify potential cost-saving options. In addition, electric-meter-based occupancy prediction systems necessitate minimum supplementary hardware or equipment, as they utilize the preexisting metering infrastructure that is already available in the majority of buildings. Electric-meter-based systems are both cost-effective and easy to implement, particularly when retrofitting existing buildings or scaling across large building portfolios. Although variations in energy use might offer valuable information about general occupancy patterns, electricity meters may not provide immediate or detailed data regarding specific occupancy trends or individual movements within the building. Dynamic environments with frequent occupancy fluctuations or temporary activities can significantly impede the precision and responsiveness of occupancy prediction systems. Furthermore, buildings with mixed-use or multifunctional spaces may pose challenges for electric meter-based occupancy prediction algorithms, as the electricity consumption may not accurately reflect the presence of people. In addition, the collection and analysis of energy consumption data may give rise to privacy problems, requiring careful consideration of data protection and security measures. Therefore, while electric-meter-based occupancy prediction offers a non-intrusive and cost-effective approach to occupancy monitoring, we should carefully assess its limitations in terms of detail and precision, taking into account specific building settings and applications.

**Table 10 sensors-24-03276-t010:** Summary of the papers in which electric meters were used.

	References	Aim of the Paper	Used Methods	Advantages	Limitations
**Electric** **mater**	[85]	Occupancy level monitoring	Statistical analysis	Non-intrusive, cost-effective, provide real- time data	Can produce false results due to weather, need knowledge for data analysis
	[86]	Room occupancy prediction	Classification algorithm		
	[87]	Household occupancy monitoring	SVM, and PCA		
	[89]	Occupancy detection and prediction	Standby state based approach		
	[90]	Occupancy detection	KNN		
	[88]	Occupancy detection	DT classification		

## 5. Data Analysis Approach

In this section, we will explain some of the main methods used in the literature to assess and forecast occupancy in buildings. In particular, in the following subsections, we will analyze analytical and ML-based methods, with a particular focus on DL algorithms. We will also explore other methods that have been used in the literature to complement the already presented ones, such as transfer learning (TL) and federated learning (FL). A top-to-down flowchart of the considered data analysis approaches is illustrated in Figure 12.

### 5.1. Analytical Methods

Analytical methods refer to a systematic approach to analyzing data and extracting meaningful features to predict future occupancy patterns. These methods involve using mathematical models and statistical techniques to understand and forecast how occupancy status in a building might change over time. They can be broadly categorized into various types, each with its own set of techniques and applications:

#### 5.1.1. Deterministic Schedule Method

The deterministic schedule method is used to predict room occupancy by developing a predetermined plan of when individuals are expected to be present in a room. These type of schedule methods have many advantages that can be used in vehicles and buildings [91]. This approach is generally used in scenarios where the room’s occupancy is known in advance, such as a classroom or a conference room. The schedule is based on various factors, such as the time of the day, the day of the week, and the planned events or activities in the room. By adhering to the schedule, it is possible to approximate the room’s occupancy at any given moment in the future. A summary of the recent papers that used this approach is shown in Table 11.

The study in [92] proposed a new population-based approach (PopAp), inspired by agent-based transportation models, to model occupant estimation. Comparing PopAp with traditional deterministic and stochastic methods, the study findings revealed notable differences in maximum occupant numbers and hourly energy demands, especially in educational buildings. These results underscore the importance of detailed occupant modeling for accurate energy system planning, particularly at the hourly scale, to effectively predict peak demand and size technologies. Another study in [93] formulated a series of occupant profiles using a deterministic method that factored in variables such as employment status, income level, household size, and age. These profiles delineated the durations occupants typically spend at home during weekdays and weekends. The dataset was subsequently classified into seven distinct profiles based on similarities, aiming to discern the predominant factors influencing residential building occupancy rates.

The deterministic schedule method is easy and stable to use for occupancy prediction, because it depends on set schedules or patterns [94]. It is very efficient with regard to computation, as it does not depend on data. However, the results achieved for occupancy prediction are not very accurate because of its assumptions. Moreover, it may also perform poorly in dynamic settings and need manual modifications to account for fluctuations in occupancy patterns or external influences.

#### 5.1.2. Stochastic Schedule Method

The stochastic schedule method is a method used to predict occupancy that considers probabilistic models and unpredictable events. This approach is used when it is impossible to forecast the exact occupancy of a space ahead of time, as in the case of a public area such as a shopping mall or airport. A probabilistic model of the events that could impact the number of people in an area is developed to forecast the occupancy status, including numbers of people. By using this model to simulate these events, it is possible to predict the occupancy of a room at any point in the future. This approach is more adaptable than the deterministic schedule method, since it accounts for uncertainties and fluctuations in occupancy. Nevertheless, a considerable amount of data and computational power are required to build an accurate probabilistic model. This method introduces flexibility by incorporating randomness and variability into projections, effectively reflecting the inherent uncertainty in occupancy patterns [95]. This flexibility enables more accurate forecasts and enhanced decision-making in ever-changing settings. Nevertheless, disadvantages to consider include the method’s complexity, heightened data requirements, and diminished interpretability in comparison to deterministic approaches. In addition, stochastic approaches may require greater processing resources and implementation time, which might pose difficulties in real-time applications. The following are some techniques used in stochastic schedule methods for the prediction of room occupancy in a building:

##### Markov Chains

A Markov chain is a mathematical method that can illustrate the progression of a system as it moves between a finite set of states. A Markov chain model was utilized to forecast the probability of a room being occupied at a specific time based on its prior state (occupied or unoccupied) [96].

Different academic and industrial applications have started to use this approach to forecast the occupancy in SBs. In the study in [97], the authors integrated various technologies, including thermal comfort experiments, occupancy simulations, usage behavior modeling, and building energy simulations. The study began by conducting human subject experiments to measure the effects of thermal comfort and occupancy prediction. The researchers used Markov chain and conditional probability models to describe room occupancy. They also employed extended comfort temperature range and user behavior models in a building energy simulation tool to analyze the energy-saving potential of a personal comfort system (PCS). Their findings indicated that using PCS can considerably enhance occupants’ thermal comfort and satisfaction in warm and cool conditions. Similarly, in another study [98], the authors examined the occupancy patterns of eight families residing in cold regions of China by collecting occupancy data from four primary rooms: the living room, bedroom, kitchen, and bathroom. The study focused on the duration of user occupancy and hourly mean occupancy, while also evaluating regular and random characteristics of the data. The researchers developed an event-based occupancy model utilizing an inhomogeneous Markov chain based on the findings. The model segmented daily events into three categories based on their randomness and included models for each room. Based on the presence probability in each time step, transition probabilities could be estimated for the three states of entering, leaving, and staying.

##### Hidden Markov Model

HMMs are an extension of Markov chains where the system being modeled is assumed to be a Markov process with unobserved (hidden) states. A HMM can predict the probability of a sequence of observations, based on a hidden sequence of states that cannot be directly observed but can be inferred from data.

Authors introduced and tested personalized and privacy-preserving collaborative-filtering models that used sequential history to predict current and future mobile context, using a combination of HMMs and DL [99]. These models could be used to verify or improve the context data obtained from sensors in real time. Similarly, researchers have developed a technique to estimate, simulate, and forecast occupant activity levels in the short term [100]. The method employs HMMs and autoregressive hidden Markov models (ARHMM). The only inputs required for the model are CO_2_ levels and the time of day. The study also demonstrated that the proposed model could simulate activity levels and corresponding CO_2_ levels. A complex environmental sensor network was implemented by Dong et al. [101] that included a wireless sensing system for environmental conditions, a wired carbon dioxide sensor system, and a wired indoor air quality sensor system. Wired cameras were used to obtain real occupancy data. The result showed that living conditions were significantly correlated with the environment. In the test, the HMM could predict room occupancy with an average accuracy of 73%. Dong and Lam later developed a Gaussian mixture model in [102]. A prediction accuracy of 83% was achieved for the number of occupants. In a classical HMM, the observed variables were assumed to be independent. During measurement of environmental parameters, such as CO_2_ concentration, there is a relationship between the number of occupants and the concentration in the previous moment. Han et al. [103] compared SVMs and hierarchical learning models. The ARHMM performed best, with an accuracy of 80.78%

There are several advantages to using HMMs. A HMM is versatile because it incorporates different types of data sources, such as sensor readings, contextual information, and environmental data. It is capable of adjusting to changes in room conditions over time, as well as to variations in occupancy patterns and dynamics within a room. This method provides probabilistic occupancy estimates instead of binary results (occupied/unoccupied), which are more informative.

In spite of the advantages of HMMs, there are some limitations as well. When dealing with large datasets, HMMs are computationally intensive. Generally, HMMs are used to predict short-term events, and they have lower performance when forecasting long-term events.

##### Bayesian Inference

Bayesian inference is a statistical method that uses prior knowledge and observed evidence to update beliefs about the probability of events [104]. This involves applying Bayes’ theorem, which describes how to revise a hypothesis’ probability as new evidence becomes available. Rather than considering probabilities as frequency estimations, Bayesian inference treats them as measures of uncertainty.

In the study in [105], a Bayesian inference approach with sensitivity analysis was proposed to analyze CO_2_ readings in four primary schools. The goal was to identify uncertainties and calibrate key parameters to estimate the probability of room occupancy based on CO_2_ readings. The study also discussed the parameters that affect the calibration performance, such as the student occupancy schedule, the number of students, and the frequency of CO_2_ readings. The results could help schools to better utilize CO_2_ meters for daily operations and interpret the readings during the COVID-19 pandemic. Similarly, another article described an applied method for conducting knowledge inference on events occurring in an SB environment, utilizing IoT technology [106]. The approach was implemented and tested in an actual SB environment, with various events inferred during evaluation, including room occupancy, elevator movements, and the conjunction of both events. Moreover, to forecast the number of occupants in a building, Ebadat et al. [107] proposed blind identification methods. The study’s authors used a Bayesian marginal likelihood method in conjunction with CO_2_ concentrations and ventilation levels to estimate the number of people in a laboratory. In experimental validation, the proposed approach achieved an accuracy of 82.1%. Finally, Yang et al. [108] developed occupancy models using ambient sensors and tree-augmented naive Bayes networks for demand-controlled HVAC systems. Binary occupancy detection algorithms, such as the NB algorithm, were 88.9–94.33% accurate, while the TAN algorithm was 95.3–98.3% accurate.

##### Markov Chain Monte Carlo

MCMC is an approach for computing complex probability distributions through repeated random sampling [109]. This technique involves generating a sequence of samples that form a Markov chain, where the current state depends solely on the previous state, utilizing these samples to approximate the desired distribution. MCMC is versatile and can be employed in various scenarios, such as forecasting occupancy in the future for SBs.

In their study [110], the authors presented a model to forecast energy consumption by occupants, based on analyzing the connection between occupant conduct and equipment energy usage. A model was created using an indoor occupancy rate and computer input power model, and polynomial and Markov chain–Monte Carlo methods were employed to depict the time-dependent indoor occupancy rate and computer input power in multi-occupant office spaces. Similarly, the paper described a practical approach to extracting knowledge about simultaneous events for automatic control in SBs [106]. The problem of the small time interval between two correlated events was resolved using the MCMC sampling method to optimize the sampling of time intervals. The proposed approach was implemented in a real SB environment and evaluated by inferring several events, such as room occupancy, elevator movement, and their conjunction. Richardson et al. illustrated stochastic occupancy using a model proposed previously in [111]. Widen et al. [112] developed a three-state occupancy model based on MCMC technology. To control lighting based on power consumption, the model was extended to nine states, and six activities involving power consumption [113]. Based on time-use data, Aerts et al. [114] identified seven significantly different indoor patterns and developed a three-state probability model for each indoor pattern. This model stored state transition probabilities in a matrix to reduce complexity. A four-state indoor activity model was developed by Mckenna et al. [115] based on Richardson et al. [111], which obtained satisfactory results.

**Table 11 sensors-24-03276-t011:** Summary of the papers using analytical approaches for occupancy prediction.

References	Used Methods	Application	Building Type	Key Findings
[92]	Deterministic Scheduling	Minimize energy	N/A	10–20% reduction in energy
[93]	Deterministic	Minimize cost, energy	SB	N/A
[97]	Markov Chains Model	Thermal comfort, energy consumption	smart room	25–40% reduction in energy
[98]	Markov Chain Model	Thermal comfort	Residential buildings	Not defined
[99]	HMM	Building/room occupancy prediction	Gym	Not defined
[100]	HMM	HVAC systems optimization	School room	Not defined
[101]	HMM	Energy consumption monitoring	smart room	Accuracy about 73%
[103]	SVM, HMM	Energy consumption reduction	smart room	Accuracy rate of 80.78%
[105]	Bayesian Inference	Air ventilation improvement	School room occupancy	Calibrated ventilation 95% fall down
[106]	Bayesian Network (BN), and MCMC	IoT-enabled sSB control	SB	Not defined
[107]	Bayesian Inference	Blind identification methods for occupancy	smart laboratory	Achieved accuracy of 82.1%
[108]	Bayesian Inference	HVAC systems efficiency	SB	NB algorithm accuracy 88.9–94.33%, TAN algorithm 95.3–98.3% accurate
[110]	Markov Chain-Monte Carlo	Energy consumption forecasting	Office space occupancy	Not defined
[111]	Markov Chain-Monte Carlo	Energy consumption forecasting	UK households	Not defined
[112]	Markov Chain-Monte Carlo	Domestic lighting demand generation	Household load forecasting	Not defined

### 5.2. Machine Learning Methods

As technology continues to evolve, the integration of IoT devices is becoming increasingly ubiquitous in our daily activities, facilitating seamless monitoring and tracking of our surroundings. This proliferation of IoT technology has resulted in the accumulation of vast data resources, which hold significant potential across multiple application domains. For instance, within building management, IoT sensors provide real-time data on variables such as temperature, lighting, and occupancy. However, manually processing such extensive datasets is both time-consuming and susceptible to errors. Consequently, there is a growing reliance on ML techniques to analyze these data efficiently, furnishing building managers with accurate insights for effective decision-making.

The following subsections introduce some ML algorithms and analyze the way they are used for occupancy detection/forecasting in buildings. Table 12 presents an illustration of the ML algorithms used across various literature articles that will be analyzed in the following.

#### 5.2.1. Support Vector Machine

An SVM is a supervised learning algorithm that can handle both linear and nonlinear relationships between features and target variables by mapping data into higher-dimensional spaces [116]. This algorithm is particularly suitable for high-dimensional datasets and can effectively handle non-linear relationships between features. To utilize an SVM for room occupancy predictions, we can train a model using a dataset of past occupancy data, which include various features such as time, day, temperature, lighting, and other environmental factors. The model will learn to identify correlations and relationships between these features and occupancy levels. After training, the SVM model can forecast real-time occupancy levels based on environmental conditions.

In the study in [64], the authors designed and tested various ML techniques, such as an SVM and a mix of classification and regression algorithms, to predict classroom occupancy. They utilized a dense wireless network to collect data from a sizeable university campus and achieved satisfactory results. Similarly, the article in [117] focused on precise forecasting on three distinct topics: identifying indoor occupancy, predicting occupancy density, and determining the exact headcount, utilizing the gathered data and applying different ML algorithms, including an SVM. The study examined how to identify the best feature set for an indoor context by properly integrating static and dynamic contexts with indoor air quality, resulting in satisfactory outcomes for both occupancy detection and prediction. Akbar et al. [89] used pattern recognition to detect occupancy states without involving intrusive methods. To illustrate the importance of selecting the right kernel function, their study compared the performance of an SVM with three different kernel functions (linear, polynomial, and radial basis functions). The accuracy of the results was between 55.37% and 79.12%, based on electricity data collected in a research center. In addition, Kleiminger et al. [87,118] investigated SVM and electricity consumption data to estimate household occupancy. It was demonstrated in these works that the selection of characteristics used in learning processes is important. Authors validated their models using the public ECO dataset1 [119] and found an occupancy detection accuracy ranging from 68 to 94 percent. According to Liu et al. [55], occupancy can be estimated using CO_2_ and temperature sensors. First, preliminary detection results were obtained using the ELM algorithm. In the second stage, a linear SVM was used to generate the final detection results. The results indicated that this approach was 97.57 percent accurate in detecting occupancy. By using WiFi, Zou et al. [120] estimated the number of occupants. An evaluation of domain-invariant kernels was conducted using transfer kernel learning. In their experiments, WiFree achieved an accuracy of 92.8 percent when counting occupancy.

For predicting room occupancy, SVMs have several advantages [121]. With their ability to handle datasets with many features, they are ideal for modeling complex occupancy patterns and relationships. An SVM is capable of modeling nonlinear relationships between features, which is important for capturing the complex interactions between environmental factors and occupancy levels. It can be combined with other ML methods, such as clustering or regression, to improve occupancy prediction accuracy.

This technique also has limitations. SVMs may perform poorly with incomplete or noisy data, which poses a challenge in occupancy prediction scenarios. Furthermore, imbalanced datasets may also affect SVM performance, especially if the number of samples varies significantly between occupied and unoccupied classes. The computational demands of SVMs can also be time-consuming and intensive, which can pose a challenge for real-time occupancy estimation in large datasets or feature spaces with high dimensions.

#### 5.2.2. Regression Method

A regression algorithm is an ML method used to model the relationship between an independent variable and a continuous variable. Based on input features, regression analysis seeks to predict the numerical value of the target variable [122]. Many studies have applied this technique to predict occupancy status; regression can forecast the number of individuals in a room based on different environmental variables like temperature, humidity, CO_2_, light intensity, and noise levels.

One work comprehensively evaluated five distinct ML techniques for occupancy detection, utilizing data from five sensor streams that were highly correlated with building occupancy [123]. A model prototype was created and subjected to training and testing to evaluate its performance. The results showed that the model performed well in predicting occupancy and showed satisfactory results. Similarly, in the article in [124], the authors described the development of a linear multi-regression model to predict the cooling load of a room in the Renewable Energy Research Laboratory at Mangosuthu University of Technology, using radiant time series method components. The model considered several predictors, including male and female occupants, window cooling load, and roof cooling load, which were identified as the most influential factors in determining the cooling load of the room.

The regression method offers both advantages and limitations. It provides flexibility, as it can predict occupancy using various variables like time of day, day of the week, temperature, humidity, and occupancy in neighboring rooms. Regression models can be effortlessly scaled to accommodate multiple floors and buildings, making it an advantageous tool for building managers managing occupancy across multiple locations. However, regression has some limitations when it comes to forecasting room occupancy. It requires a significant amount of data to be effective, and if data are not available or of poor quality, this may lead to inaccurate predictions. The regression method may not be applicable in all settings, such as open-plan offices or spaces where occupancy is difficult to measure or predict.

#### 5.2.3. K-Nearest Neighbors (KNNs)

KNNs is a ML algorithm used to predict classifications or regressions of new data points by finding the *k* closest neighbors in a feature space and using either their majority class (for classification) or average value (for regression) [125]. It can predict whether a room is vacant or occupied based on data collected by sensors such as motion, door, temperature, and light sensors. To employ KNNs for occupancy prediction, a dataset of historical sensor data is first gathered. The dataset is then split into training and testing sets, with the training set used to train the KNN model and the testing set used to assess its accuracy. The K value, which specifies the number of nearest neighbors to consider, is selected based on the dataset and the desired level of accuracy. Once the KNN model has been trained and tested, it can be applied to real-time data collected by sensors to predict room occupancy [70].

The study in [126] involved comparing multiple methods for predicting the number of indoor users, with and without using SB state variables. The experiment collected data on indoor temperature and CO_2_, as well as device log data, in a SB. The study evaluated the effectiveness of five ML techniques, including KNN, GP, RF, BR, and MLP. Similarly, in another paper, the authors introduced a novel, cost-effective, and eco-friendly method for detecting occupants in enclosed spaces using passive cognitive radio (CR) [127]. The proposed solution utilized a reconfigurable software-defined radio system and adaptive spectrum sensing technology. The experimental results demonstrated that the CRhodora system could effectively and accurately detect human occupancy in indoor spaces. Additionally, Based on the research of Vela et al. [128], KNNs was selected as the most effective occupancy estimation algorithm. Nevertheless, its use must be considered because of its low complexity. Additionally, this algorithm has a shortcoming in terms of classification. When the sample is unbalanced, for example, if one class has a large sample size and others have a small sample size, this may cause confusion among the K neighbors of the sample when a new sample is input, particularly if large-volume samples dominate.

KNNs is a simple and flexible method for predicting building occupancy [121]. Because of its ability to learn locally, KNNs is more applicable in real-world scenarios and can adapt to changing occupancy patterns. KNNs does have limitations though. With large datasets in particular, the computational requirements can be high. For maximum performance, it is also essential to choose the ideal value for the hyperparameter K.

**Table 12 sensors-24-03276-t012:** Summary of the papers using ML approaches for occupancy prediction.

References	Used Methods	Application	Building Type	Key Findings
[64]	ML methods	optimize the usage of classroom	university campus	accuracy of almost 84.6%
[117]	SVM	sensor fusion to enhance accuracy	Smart room	99.85% for occupancy detection, 92.9% for occupancy estimation
[118]	ML, SVM	used electric meter as a occupancy sensor	household residence	80% improvement
[55]	Extreme learning machine (ELM)	energy consumption management	smart room	accuracy is 97.29%
[120]	WiFi based device-free occupancy detection	HVAC systems	smart room	99.1% for occupancy detection, 92.8% for occupancy estimation
[123]	Regression	thermal comfort	SB	99.7% for occupancy detection, 99.3% for occupancy estimation
[124]	linear regression	HVAC system control	SB	relative errors are 0.0073, 0.0016, 0.0168 and 0.0162%
[126]	KNN ML	HVAC system control	smart room	Mean Absolute Error (MAE) = 0.036
[127]	KNN ML	energy consumption	enclosed spaces	accuracy about 97.2%
[128]	SVM, HMM	HVAC system control, lighting, security and emergency	building occupancy	accuracy rate of 97%

### 5.3. Deep Learning

Among ML algorithms, artificial neural networks (or simply neural networks) represent a disruptive technology that has revolutionized various fields by enabling complex pattern recognition, nonlinear relationship modeling, and high-dimensional data analysis. A particular subset of neural networks is represented by DL. DL uses multiple layers of interconnected neurons to extract hierarchical features from data, to enhance its ability to tackle complex tasks and learn intricate patterns with respect to traditional ML [129]. As these neural networks are capable of autonomously detecting and comprehending features from data, they are particularly beneficial for tasks that involve large amounts of data and complex relationships. In the following subsections, we will introduce some DL algorithms and analyze how they can be used for occupancy prediction in buildings. An overview of DL algorithms found in various literature articles can be found in Table 13.

#### 5.3.1. Convolutional Neural Networks

A CNN is a DL model that learns hierarchical representations of input data through convolution and pooling operations, which allows it to capture more complex features at multiple levels of abstraction [130].

Numerous studies in the literature have employed CNNs to predict occupancy in buildings, and this approach is gaining popularity. One study utilized six different meta-models [131], which included both ML and DL techniques, to predict the hourly performance of gymnasiums based on various design parameters and varying weather conditions. The meta-models were trained and tested using a large dataset generated by Energy Plus simulations for four gymnasium located in other cities in China. The study evaluated each model’s accuracy, efficiency, ease of use, robustness, and interpretability. Similarly, another paper introduced a method called CNN-XGBoost [36] for detecting occupancy in residential buildings with a balanced mechanical ventilation system using indoor climate sensors. The technique used a simple deep-learning model and inexpensive sensors to detect occupancy. Unlike previous methods that required testing in a specific room with restrictions on the use of doors, windows, HVAC, etc., the proposed method was validated in a single-family residential building, without imposing any such restrictions. Moreover, in their paper, Bao et al. [132] proposed a new method of counting people with a CNN using IR-UWB radars with a low radiation dose. According to their study, this algorithm performed better for scenes with a wider detection angle and a larger detection range when obstruction or superposition occurred. According to Tang et al. [133], CNNs were used to train occupancy sensing models, which provided good estimation accuracy. According to Conti et al. [134] two algorithms were proposed to count people in a classroom based on CNNs. Based on their results, they achieved very good results regarding people counting. In the early days, many studies were focused on detecting bodies or heads, and others were based on mapping local or global features to actual numbers. Recently, it has been proposed that the population-counting problem cab be solved using a regression of a density image of the population, with the number of people in the image being calculated by summing the values. This method can handle severe crowd occlusions. The success of DL technology has allowed researchers to use CNNs to generate more accurate population density images than traditional methods. It has been demonstrated that CNNs can be applied to people counting in complex scenes, with good results [135,136].

The use of CNNs for occupancy prediction in buildings has both advantages and limitations [13]. CNNs are highly effective at capturing and analyzing the spatial relationships present in data. This allows them to expertly examine spatial trends in building layouts and occupancy distributions, resulting in more precise forecasts. CNNs provide the ability to autonomously acquire pertinent characteristics from input data, hence diminishing the necessity for manual feature engineering. This capability allows adjusting to various building configurations and occupancy patterns, without requiring a lot of preprocessing. CNN models are well-suited for implementation in buildings with a high number of occupancy sensors or IoT devices, as they can effectively handle substantial amounts of data after training. While there is no doubt that CNNs offer several advantages, they also carry some limitations. CNNs typically require large amounts of labeled data for training, which may be challenging to obtain for occupancy prediction in buildings, especially in diverse environments with varying occupancy patterns and building layouts. Training and deploying CNN models requires significant computational resources, including high-performance GPUs and memory.

#### 5.3.2. Recurrent Neural Network

A RNN is a type of artificial neural network that was designed for the purpose of efficiently processing sequential data by maintaining an internal state or memory [121].

According to Wang et al. [70], WiFi probe technology was used to intentionally examine the requests and responses made between the access points and network devices of building occupants. The authors suggested using a Markov-based recurrent neural network (M-FRNN) to model and forecast presence patterns using collected data. Based on temperature and/or likely heat source information, the authors of [137] developed an algorithm that used support vector regression and RNN algorithms to detect occupancy behavior in buildings. Based on a limited number of wireless packets, Billah and Campbell [138] proposed a system for estimating area occupancy with a fast gated recurrent neural network (FastGRNN) operating on a BLE device. This provided energy-efficient real-time analytics.

In the context of predicting occupancy in buildings, RNNs offer both advantages and limitations [121]. An RNN provides a unique benefit in that it considers past occupancy levels, which enables it to recognize temporal patterns and forecast future occupancy levels more accurately. Furthermore, they are adept at learning long-term the dependency between past and future occupancy, which helps provide more accurate and reliable estimates of room occupancy. Considering the vanishing gradient problem caused by recurrent elements between layers, RNNs can have trouble learning long-term dependencies. As they were designed to work with text or sequential data, they are limited in their ability to process more intricate information, such as images or videos.

#### 5.3.3. Long Short-Term Memory Networks

A long short-term memory (LSTM) network is a type of RNN designed to accommodate long-term dependencies in sequential data and overcome the vanishing gradient problem. An LSTM network is made up of memory cells that have self-regulating gates, allowing them to selectively remember or forget information over time [17].

In the literature, LSTM has been widely used for occupancy prediction. The study in [139] introduced a method for predicting occupancy in smart homes, which relied on environmental factors like CO_2_ levels, noise, and temperature, and employed a ML method and a forecasting strategy. The proposed algorithms aimed to improve the energy management system by optimizing the use of the electric heating system. The study utilized an LSTM neural network. Moreover, the authors demonstrated that FL can be utilized to improve occupancy predictions when training a specific model is not feasible. The study highlighted how this approach can be applied to rooms where the occupancy patterns are unknown and, therefore, a custom model cannot be trained. Using the concept of occupancy detection without covering the whole room with sensors was proposed by Husnain and Choe in [140]. A decision module predicted human presence patterns using LSTM for the sensor’s off-range region. This reduced the installation costs for occupancy detection systems. On the same tangent, Pešić et al. [141] proposed a technique for detecting occupancy utilizing a fusion of WiFi and Bluetooth data and a set of data analytics functions to examine occupancy data across logical and physical boundaries. An LSTM neural network was studied for occupancy forecasting, and the same data analytic features were used to present and predict occupancy statistics. For workdays, they achieved 75.45% similarity on real (EDR) signals. Chang et al. in [142] used six forecasting models to analyze the same dataset: Gaussian process regression, regression by least squares, regression by backpropagation, regression by general regression, and regression by LSTM. The numerical results demonstrated that LSTM networks were superior to the other models in estimating hotel accuracy rates across three data repositories. Regarding root mean square error (RMSE), the model achieved a value of 13.31%. According to Elkhoukhi et al. [143], their main objective was to evaluate the accuracy of forecasting occupant numbers using contemporary DL methods, including a RNN and LSTM. Hitimana et al. [144] used a multivariate time series to predict occupancy patterns in regression forecasting. An empirical evaluation showed that the designed solution effectively collected, processed, and stored environmental data. LSTM was used to model the acquired data and then compared to various ML techniques, showing good performance.

There are several advantages to the use of LSTM networks over other ML algorithms [19]. A key advantage of LSTM is its ability to handle time series data over a long period, which is crucial for predicting room occupancy patterns as they change over time. Additionally, LSTM can capture the dependencies between past, present, and future occupancy patterns. In addition to identifying and capturing nonlinear relationships between various features and occupancy patterns, LSTM is able to capture multiple factors that can influence occupancy patterns, including weather, time of the day, or day of the week.

In spite of the benefits of LSTM networks for predicting room occupancy, they also have a few limitations. To be effective, LSTM models often require a large amount of occupancy data and can become too complicated. In particular, it may be problematic if the occupancy data are inadequate or incomplete. Furthermore, LSTM models are prone to overfitting the training data, leading to poor predictions when new or unseen occupancy patterns emerge.

**Table 13 sensors-24-03276-t013:** Summary of the papers using DL approaches for occupancy prediction.

References	Used Methods	Application	Building Type	Key Findings
[131]	ML and DL techniques	predict multi-performance vectors of gymnasiums	gymnasium occupancy	0.993, 0.982 and 0.941 for energy, temperature, and CO_2_ respectively
[132]	DL	occupancy estimation	residential buildings	accuracy is 92%
[133]	CNN	occupancy detection and estimation	SB	accuracy of 99.53% occupancy detection and 98.14% estimation
[136]	CNN	energy consumption management	SB	6.70% energy saved
[145]	RNN, LSTM	control building ventilation	SB	Accuracy is 95%
[70]	ANN, KNN, SVM	building energy efficiency	SB	RMSE is 2.7 for ANN
[137]	SVM and RNN	HVAC system control	SB	produced a 0.638 average error and 5.32% error rate
[138]	fast gated recurrent neural network	building occupancy detection	SB	Not defined
[7]	LSTM	energy consumption in a building	smart office	accuracy about 94.2%
[139]	LSTM	Energy management system	SB	accuracy rate of 99.16%
[140]	LSTM	SB system	smart room	accuracy rate of 95.62%
[142]	LSTM	lighting system in a building	SB	RMSE is 13.31%
[143]	LSTM	energy efficient buildings	SB	accuracy is 70%
[144]	LSTM	occupancy prediction in building	SB	model’s accuracy is 96%

### 5.4. Other Methods

In this section, we will investigate additional strategies from the literature that have been used for occupancy prediction. In particular, we will explore TL and FL. These new research topics offer distinct advantages and are demonstrating their potential for improving the predictive capabilities of the models introduced in the sections above.

#### 5.4.1. Transfer Learning

TL refers to the process of using knowledge obtained from a preexisting model trained on large datasets in order to improve the performance of a model trained on a smaller dataset [146]. The process involves transferring knowledge between domains or tasks. In general, (i) TL helps save computational resources and improve efficiency when training new DL models, since the latter can be pre-trained offline on large-scale datasets and then improved on small datasets; (ii) a DL model is trained on an annotated dataset and then validated on an unlabeled [147] dataset, which is a challenging and time-consuming task considering that data labeling is time-consuming and requires expert intervention; (iii) TL can use simulated or synthetic data rather than real-world environments to train DL models [148]; and (iv) TL can leverage knowledge gained from previous campaigns to improve generalization to other data [149]. The study in [150] proposed a TL approach as a solution to common issues that arise when implementing ML in buildings, such as adapting a model to a new building, gathering the necessary training data, and ensuring the model’s robustness in changing conditions. The study specifically focused on the practical application of a DL model for predicting room occupancy using indoor climate IoT sensors.

The TL technique has many advantages, as well as some limitations [151]. With TL, pre-trained models can be easily applied to smaller target datasets, thereby improving their accuracy and robustness. This reduces the time and expense of collecting ground truth data for training the model. A further advantage of TL is that it allows the use of learned features that are common across various datasets to scale the model across different buildings or environments. The transfer of learning may be limited in some ways. There is the possibility that the pre-trained model used in TL is not appropriate for the particular target dataset or environment, which results in suboptimal performance and the need for further fine-tuning or customization. It is possible that the target dataset does not have enough similarities to the pre-trained dataset, which will limit the effectiveness of the transferred knowledge. It may be difficult for some organizations or individuals to implement and fine-tune TL models, due to the requirement for significant computing resources and expertise.

#### 5.4.2. Federated Learning

The concept of FL involves using multiple devices to train a model, without having to share raw data with a central server [3,152]. Methods like this are beneficial when sensitive data, such as financial or healthcare records, cannot be shared. FL allows models to be trained from multiple data sources, to predict the occupancy of a room without requiring centralized data collection. The model can be used to build a global model of occupancy levels by merging local models from each device with the data remaining on the device. FL is very important for taking advantage of the so-called edge-cloud continuum, which allows for simple distribution of computing loads to edge, fog, and cloud layers [153,154].

The authors of [7] proposed an approach based on FL for the estimation of room occupancy. The paper had two main objectives. Firstly, it proposed a method for occupancy prediction in multiple rooms of a building using FL and LSTM neural networks. Secondly, it aimed to demonstrate how FL can assist in occupancy prediction for spaces where a dedicated model has not been trained. The effectiveness of the proposed approach was demonstrated through simulation experiments. Similarly, the paper in [155] focused on joint resource allocation for human motion recognition using wireless sensing in ambient intelligence. The authors began by examining the wireless sensing process and discovered a threshold value for the sensing transmit power that provided satisfactory sensing of data samples. The paper then proposed a solution to the joint resource allocation problem, taking into account constraints such as training time, energy supply, and sensing quality for each edge device. Finally, the paper in [156] introduced a multi-agent reinforcement learning framework for a joint energy and carbon allowance trading mechanism in a building community. The proposed approach included an FL technique to accelerate the training process and protect the privacy of individual building data.

Several advantages of FL should be considered [3]. Among its key advantages is its ability to perform distributed learning without sharing sensitive data. By allowing data to be accessed from multiple sources without the need to centralize or transfer, FL can also help overcome limitations related to data storage and availability. When using FL, we can combine knowledge from multiple sources and leverage the diversity of data across different environments. As FL allows for parallel training on multiple devices, it can be more efficient in terms of computation and training time.

FL has the disadvantage of requiring a large number of devices to participate, to ensure that training data are diverse and unbiased. Performance may not be satisfactory in the case of only a few devices. Additionally, communication between the devices and the central server can be slow or unreliable, causing training delays and decreased system efficiency.

## 6. Discussion, Challenges, and Future Direction

Occupancy prediction is an effective tool for mitigating energy consumption within buildings, since they consume nearly half of all energy consumed globally. However, occupancy prediction in buildings faces numerous challenges.

Selecting appropriate sensors from the market that can collect occupancy-relevant environmental data is a significant challenge. Choosing the optimal sensor is a difficult task, compounded by the fact that one sensor may not be sufficient. Therefore, identifying the right sensor combination is another challenge. It is also important to consider the placement of sensors when detecting and predicting occupancy. Choosing the right locations for sensors within a building is crucial for capturing accurate occupancy data. Sensor placement errors can result in incomplete or biased data, which can lead to inaccurate predictions. It is also important to keep in mind that the cost of sensors can be a significant barrier to adoption. Due to the high cost of some sensors, it is essential to determine the minimum number of sensors needed to achieve reliable predictions, while minimizing costs. Implementing occupancy prediction systems involves balancing the need for sufficient data coverage with cost considerations. The process of storing and processing the data collected from these sensors also raises concerns regarding privacy. By using occupancy sensors, we may be able to collect sensitive information about the behaviors and routines of our building’s occupants. Thus, it is paramount to ensure the privacy and security of these data.

Following the collection of data by sensors, the next challenge is to decide which methodology will be most appropriate for processing the data and making informed predictions regarding occupancy. Choosing the right processing method is crucial, since it directly impacts the accuracy and reliability of predictions. A challenge in forecasting occupancy levels is fine-tuning the parameters of the chosen method, which requires precision adjustments.

In this field, another challenge pertains to selecting the optimal physical location for data collection and algorithm execution. Several studies in the literature investigated the efficacy of edge computing/intelligence [157,158] in decentralizing computation to create more responsive environments and to maintain the proximity of data to their source. Additionally, there has been a growing trend in recent years towards leveraging the device–edge–cloud continuum [154], enabling algorithms to be executed where they are most needed and feasible.

The following subsections will explore future research directions to improve overall energy efficiency, occupant comfort, robust occupancy prediction, and privacy within SB environments.

### 6.1. Block-Chain Based IoT SB Environments

As an emerging technology, the integration of blockchain technology with the IoT for occupancy prediction in SBs has great potential for future development [159]. Blockchain-based IoT occupancy prediction has the potential to improve security and privacy, integrate with smart contracts, improve data quality and accuracy, integrate with artificial intelligence and ML, and be scalable and interoperable with other SB technologies in the future [160,161]. While it is important to recognize the significant costs associated with blockchain technology, it is equally important to underscore its potential in creating secure and energy-efficient buildings [162]. By harnessing blockchain’s cryptographic principles, sensitive occupancy data can be securely stored and accessed, instilling greater trust among stakeholders and promoting increased data sharing. This aspect is particularly noteworthy in today’s digital landscape, where individuals prioritize the protection of their personal data and are wary of its misuse. By ensuring the privacy and security of occupancy data, blockchain enables the utilization of a larger and more comprehensive dataset. Consequently, this enhances the accuracy of occupancy predictions and empowers more precise energy-management strategies. In essence, the integration of blockchain technology can not only address concerns regarding data privacy but also play a pivotal role in optimizing energy efficiency within SBs.

### 6.2. Sensor Fusion and Optimal Sensor Placement

Predicting room occupancy in an SB environment involves the fusion of sensors and the optimal placement of sensors [10,163]. The concept of sensor fusion refers to the integration of multiple sensors that measure various aspects of the environment, such as temperature, humidity, light, and motion, to better understand the occupancy of the room. The occupancy of a room can be more accurately determined by combining data from multiple sensors.

Sensor fusion in SB environments is indeed a powerful tool for enhancing room occupancy prediction accuracy and overall system performance. However, it is essential to weigh this against the associated cost implications. While combining sensors can provide comprehensive data insights, it also escalates the expenses involved in deploying and maintaining a sensor network.

Researchers have recognized the need to strike a balance between accuracy and cost-effectiveness in sensor deployment strategies. Some studies have focused on identifying the minimum number of sensors required to capture essential environmental parameters, without compromising prediction accuracy [47]. Nevertheless, even if individual sensors are relatively inexpensive, the cumulative cost of deploying multiple sensors across a building can still be substantial.

As such, future research endeavors could delve deeper into optimizing sensor placement and selection methodologies, to minimize costs, while maximizing the efficacy of room occupancy prediction systems. This would entail exploring innovative sensor configurations and leveraging advanced data analytics techniques to extract meaningful insights from limited sensor inputs. By addressing the dual objectives of accuracy and affordability, researchers can pave the way for more sustainable and accessible SB solutions.

### 6.3. Unsupervised Learning

Typically, occupancy prediction methods require a set of ground truth occupancy data, which can be difficult and costly to collect, making their use difficult in practice. In the last few years, there has been increased interest in exploring unsupervised learning methods as potential alternatives for detecting and predicting occupancy [164]. Data can be learned without labeled data, making unsupervised methods suitable for situations without ground truth data. Unsupervised methods have been explored for occupancy detection, including clustering, anomaly detection, and neural networks [165]. As an example, the paper in [166] introduced an unsupervised ML method based on finite mixture models. The study used a method called scaled Dirichlet distributions, known for their flexibility and efficiency in various applications. The proposed algorithm employed entropy-based variational Bayesian inference to learn finite scaled Dirichlet mixture models. The results were compared with existing methods to demonstrate the efficiency of the proposed framework. Although unsupervised methods are not as accurate as supervised methods, these methods are promising as a means of developing more cost-effective and practical solutions for occupancy detection in SBs. As part of future work, it may be possible to investigate this area for predicting occupancy in SB environments.

### 6.4. Activity Forecasting

There is no doubt that occupancy prediction plays a crucial role in improving the efficiency of SBs, as it enables control of HVAC systems, lighting, and computers, to make them more efficient. We can, however, extend the capabilities of SBs and occupancy prediction by integrating activity recognition and forecasting, to achieve more than just controlling the above-mentioned devices. With this advancement, multimedia devices can be controlled in office buildings and household appliances can be controlled in homes. Moreover, all the systems in the buildings can be tuned, not only on the forecast presence of people, but also on their specific activities.

Numerous studies have been conducted on the topic of activity recognition within SBs [167]. Traditional methods of activity recognition can raise privacy concerns for building occupants, especially when cameras or similar technologies are used [168]. As a result, researchers have begun using advanced technologies such as UWB radar for the detection of activity within buildings [27]. The adoption of UWB radar technology for activity recognition in SBs shows promise for enhancing energy efficiency, occupant comfort, and privacy. However, challenges such as generalization issues and sensitivity to changes in radar positioning should not be ignored. Overcoming these challenges requires developing more adaptable ML algorithms and advancing radar hardware and signal processing techniques. Once these hurdles have been addressed, incorporating FL could further enhance data privacy, while optimizing building operations and occupant experiences. While this research area is relatively new and not yet mature, it holds great potential for future research directions intended at enhancing energy efficiency, occupant comfort, and privacy within SB environments.

### 6.5. Occupant Localization

While both occupancy prediction and activity recognition are critical for optimizing energy efficiency within SBs, the importance of occupant localization cannot be ignored [169]. Occupancy localization allows us to track an occupant’s location inside a building. Using this information, we can deactivate appliances in areas where there are no occupants and activate them when there are occupants nearby. There have been numerous studies conducted that used this technique to improve the energy efficiency of buildings [170]. Nevertheless, there are gaps for further exploration, such as the control of complex scenarios, the scalability to the number of occupants, and the management of changes in the environment in which localization is running [170]. Occupant localization is essential for optimizing energy efficiency in SBs, allowing targeted control of appliances based on occupant presence [169]. While increasing the number of sensors or radar can improve precision, this raises cost considerations. Further exploration is needed to balance performance and cost-effectiveness. Challenges include adapting to complex scenarios, scaling to accommodate more occupants, and managing environmental changes. Future research should focus on optimizing sensor placement, leveraging advanced data fusion techniques, and exploring ML algorithms tailored for localization tasks, to enhance system performance and scalability. Addressing these challenges will lead to more efficient and cost-effective SB solutions.

## 7. Conclusions

This paper presented a review of several papers concerning occupancy prediction within smart building environments. The study primarily focused on delineating various monitoring technologies applicable to indoor settings, encompassing mobility sensors, non-mobility sensors, cameras, radars, and smart meters. It highlighted that each of these technologies carries distinct advantages and limitations, contributing to a nuanced comprehension of their suitability for occupancy prediction scenarios.

Moreover, this paper delved into the diverse methodologies utilized for data processing, encompassing analytical methods, ML algorithms, and various DL architectures. Each technique was further divided into sub-branches, offering detailed explanations to enhance readers’ grasp of the computational methodologies employed in occupancy prediction models. Additionally, some other methodologies, such as federated and transfer learning, were explored for their potential to enhance the privacy of model training data, as well as to improve scalability and performance.

Furthermore, we discussed the challenges associated with occupancy prediction, particularly regarding data gathering, data analysis, and the place in which algorithms can be executed. Lastly, this paper contemplated potential future directions and proposed approaches to overcome the aforementioned challenges.

## Figures and Tables

**Figure 1 sensors-24-03276-f001:**
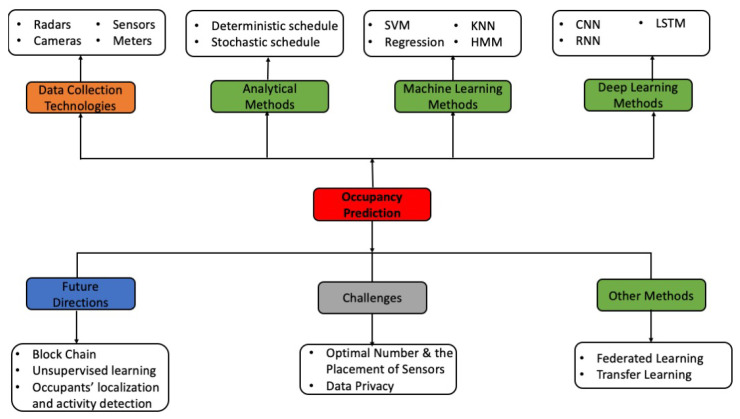
Overview of our study.

**Figure 2 sensors-24-03276-f002:**
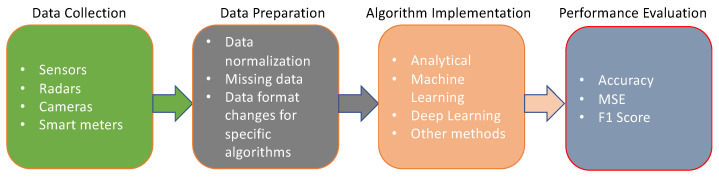
Overall flowchart of the occupancy prediction process.

**Figure 3 sensors-24-03276-f003:**
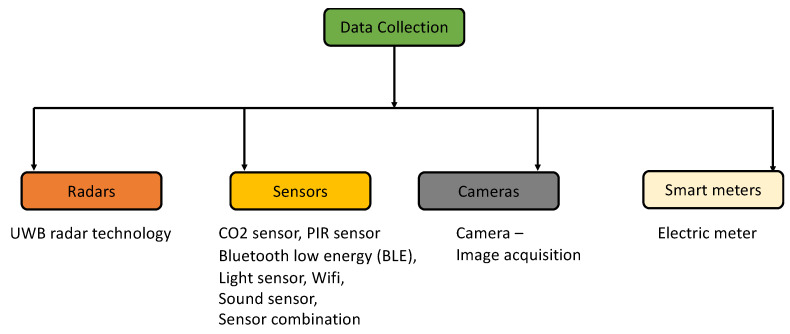
Set of data collection techniques used for monitoring occupancy environment.

**Figure 4 sensors-24-03276-f004:**
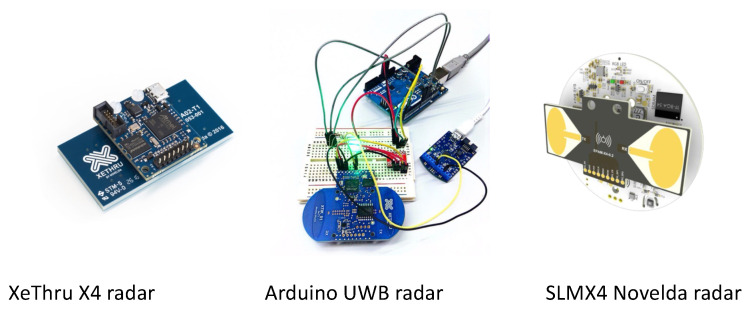
Some example devices of UWB radar used for occupancy detection, estimation, and prediction.

**Figure 5 sensors-24-03276-f005:**
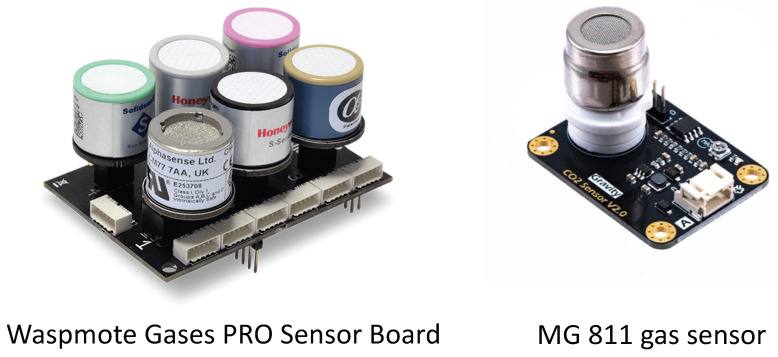
Some example devices of CO_2_ sensors used for occupancy detection, estimation, and prediction.

**Figure 6 sensors-24-03276-f006:**
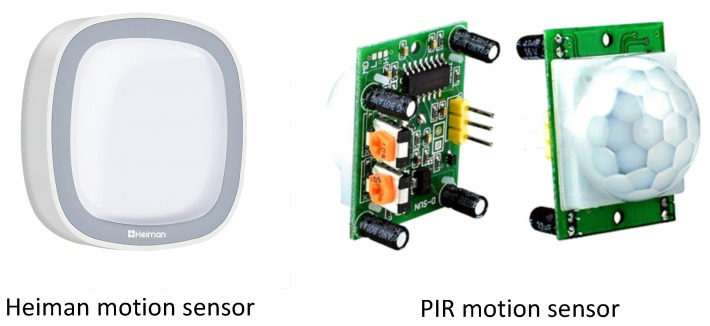
Some example devices of PIR sensors used for occupancy detection, estimation, and prediction.

**Figure 7 sensors-24-03276-f007:**
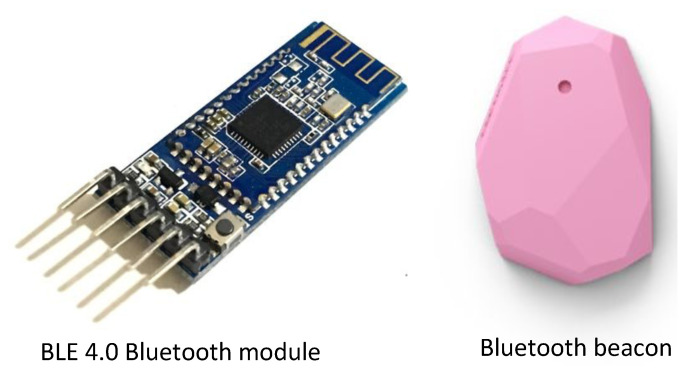
Some example devices of Bluetooth technologies used for occupancy detection, estimation, and prediction.

**Figure 8 sensors-24-03276-f008:**
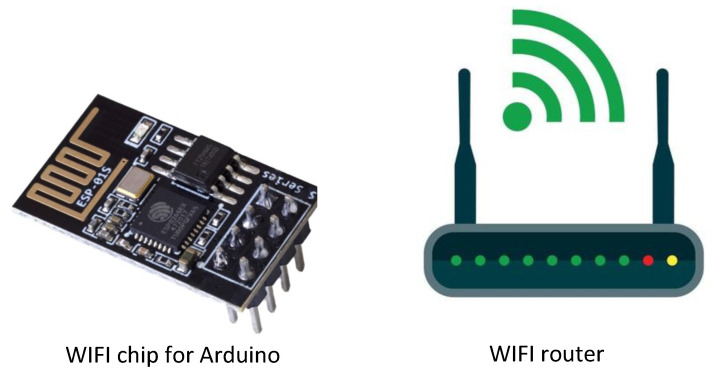
Some example devices of WiFi technologies used for occupancy detection, estimation, and prediction.

**Figure 9 sensors-24-03276-f009:**
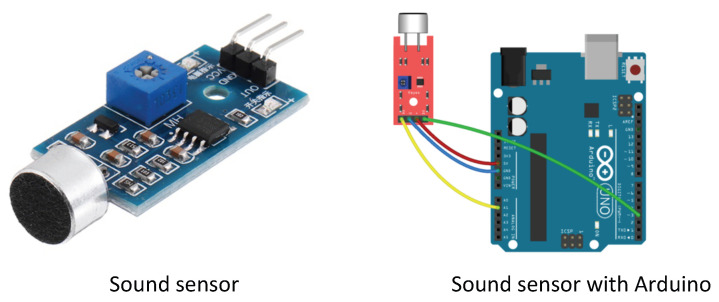
Some example devices of sound sensors used for occupancy detection, estimation, and prediction.

**Figure 10 sensors-24-03276-f010:**
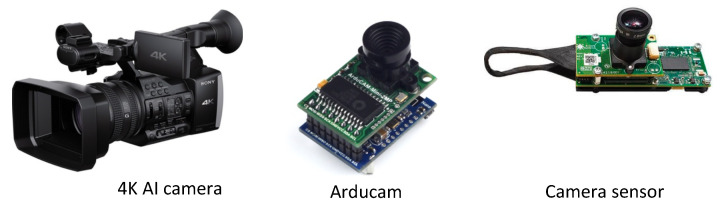
Some example devices of cameras technologies used for occupancy detection, estimation, and prediction.

**Figure 11 sensors-24-03276-f011:**
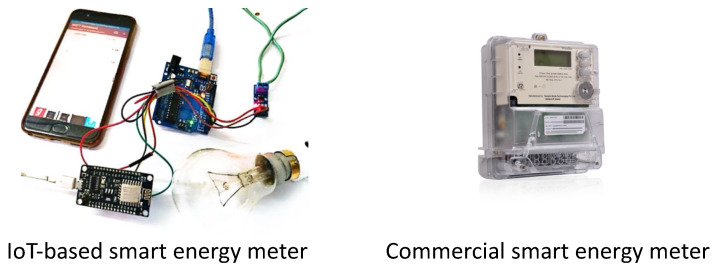
Some example devices of smart energy meters used for occupancy detection, estimation, and prediction.

**Figure 12 sensors-24-03276-f012:**
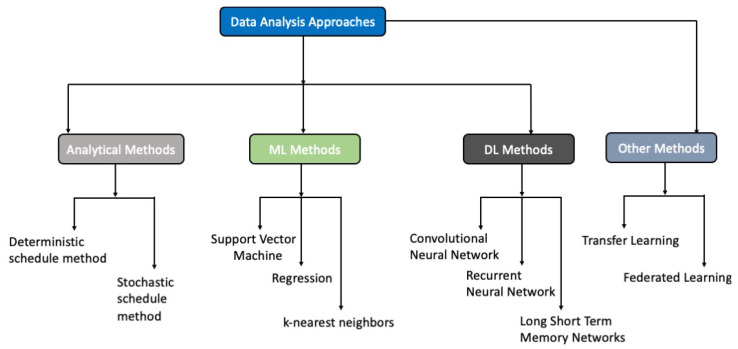
A top-down flowchart of the considered data analysis approaches.

## Data Availability

Data are contained within the article.

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
