# Peer review of "Occupancy Prediction in IoT-Enabled Smart Buildings: Technologies, Methods, and Future Directions"

_sensors, 2024, doi:10.3390/s24113276_

Round 1

Reviewer 1 Report

Comments and Suggestions for Authors

The paper presents a valuable survey of work related to occupancy prediction in cognitive buildings revising an impressive number of publications from different repositories and covering many technologies and algorithms. The authors classified the work in data acquisition technologies, algorithms for the prediction, and open issues. This classification is presented in Sections 2, 3, and 4. I have some comments on these sections, as detailed in what follows.

Considering the data acquisition technologies, the authors present a classification based on the type of sensors that can be used. However, when presenting the technologies the analysis concentrates on the algorithms used for the predictions rather than the pros and cons of each one, which are briefly introduced at the beginning of each section. This approach does not help in guiding the reader in understanding the benefits of each technology and its impact on the prediction algorithms that can be used. Moreover, the prediction algorithms are supposed to be the topic of another section of the paper (Section 3). I would recommend clarifying this part and providing a more in-depth analysis of the considered technologies from this perspective. This approach is also adopted in all the tables in Section 2, where all of them provide some general advantages and limitations of each technology without differentiating with the method used, highlighting the fact that the methods seem not correlated with the technology adopted. Thus, I would recommend presenting a correlation or any taxonomy that can correlate methods with the technologies or leave the methods for their dedicated section.

In Section 2.5.3, [69] presents a hybrid solution, where sound sensors are not used alone, whereas sensor combinations have been already presented in Section 2.4.

In Section 2.5.4, I would recommend to include some analysis related to the privacy issues.

In Section 3, I would recommend using quantitative terminology when referring to the accuracy (i.e., avoiding the usage of good, bad, etc.) to allow the reader to compare the different presented solutions. Similarly, when referring to the data needed by each solution, I would recommend using quantitative terminology (i.e., no “considerable/significative/large amount of”).

Section 3.3.1 presents many works on CNN, I would suggest introducing some taxonomy concerning the data used to help the reader understand the pros and cons of CNN in different scenarios.

In Section 3.3.2, [131] already introduces LSTM, which is supposed to be analyzed in Section 3.3.3. Similarly, FL is supposed to be analyzed in Section 3.4.2 but it is already introduced in Section 3.3.3.

In Section 4, the authors affirm that it is important to correctly select the technology to gather data and then the method to process it; however, the paper mixes these two aspects and does not provide clear guidelines for these two choices. I would recommend providing more clearer analysis in this respect and also some hints on the correlation of the two selection processes, if any.

In Section 4.1, I would recommend providing references and a clearer explanation of the capacity of BC to improve data quality, accuracy, and energy efficiency (considering that many papers highlight the high energy cost of BC).

In Section 4.3, I would recommend to extend a but the analysis by providing some examples of unsupervised ML solutions that can be adopted in SB/CB scenarios. In general, all these sections on future directions would deserve a more clear explanation about the motivations and the possible methodologies that can be used.

In conclusion, the two sections about technologies and methodologies seem to mix their content, which makes it difficult to have some general conclusions. The sections related to future directions lack details and deserve a more in-depth explanation. 

Comments on the Quality of English Language

In general, the English is good. However, several acronyms are not introduced the first time they are used, they are not introduced at all, or they are introduced multiple times. I would recommend double-checking all acronym definitions.

Author Response

Please see the document attached.

Reviewer 2 Report

Comments and Suggestions for Authors

The paper "Occupancy Prediction in IoT-enabled Smart Buildings: Technologies, Methods, and Future Directions" explores the use of Internet of Things (IoT) technologies to enhance energy efficiency in buildings by accurately detecting, learning, and predicting occupancy patterns. It addresses the problem of energy wastage due to inefficient use of electrical appliances such as HVAC and lighting systems. By making buildings smarter and more cognitive, they can adjust their operations based on the presence of people, leveraging advanced technologies and methods to sense occupancy. The paper concludes by discussing potential future directions to improve occupancy prediction capabilities in smart building environments, aiming to reduce energy wastage and increase sustainability. Authors are advised to address the following comments in the revised version.

1-    Improve of the abstract to better articulate the study's innovation, methodology, research findings, and significance. A clear and concise abstract will help to immediately convey the value and novelty of the research to readers.

2-    Write key objectives of this study in 3 points at the end of the introduction section.

3-    Write paper organization at the end of the introduction section.

4-    Extensive literature review is needed. A detailed comparison table of the existing schemes must be added in a new Section 2. Also, authors need to discuss the following IoT studies with their strength and weakness in a comparison Table 1: - A comprehensive survey on the cooperation of fog computing paradigm-based iot applications: layered architecture, real-time security issues, and solutions." IEEE Access (2023).  -Blockchain and Internet of Things in smart cities and drug supply management: Open issues, opportunities, and future directions. -A lightweight smart contracts framework for blockchain‐based secure communication in smart grid applications." IET Generation, Transmission & Distribution (2024).

5-    A data collection methodology figure should be added in Section 3. Detail must be discussed in the text.

6-    Add a related figures and explanation in each subsection of Section 2.

7-    A flow chart for data analysis method or framework must be added in Section 3.

8-    A top to bottom figure of Machine learning algorithms must be added in Section 3.2

9-    Create a new section and discuss the existing issues, potential solutions, and discussion in detail.

10-  Provide the conclusion section that concludes the entire scheme in detail. 

Comments on the Quality of English Language

Minor

Author Response

Please see the document attached.

Reviewer 3 Report

Comments and Suggestions for Authors

-The paper provides a very good and thorough review

-The abstract reads more like an introduction. I suggest that you reduce it a bit and include more of the outcomes of the research inside the abstract. Currently there is very little in terms of outcomes of the research in the abstract and every abstract should include part of the conclusions/results.

-line95: UWB is not defined anywhere in the text before this point. Define it when it is first encountered in the text

-Section 1.1 is good and table 1 provides a good overview

-The objectives of the research are not clearly articulated. This should be done before the methodology part. In the methodology part, the methods used to meet these objectives should be stated and at the end of the paper there should be a short discussion whether the objectives were met.

The introduction section could be a place where the aim and objectives of the research can be stated.

Currently, there is only one paragraph at the end of the introduction that only identifies the gaps in research. Then it goes into a literature review and somehow the aim and objectives of the research are left not defined.

-Tables 2 -10 are very good, but they can also be a bit more in depth. Some more in-depth insights from these references can be provided in these tables in a per paper basis (i.e. Potentially add 1 more column with significant individual findings per paper placed there) and keep all other columns as well. Currently, the information provided at the se tables is quite high level.

Author Response

Please see the document attached.

Reviewer 4 Report

Comments and Suggestions for Authors

The authord proposed an interesting survey on method and technologies for building occupancy prediction an detection.

The manuscript is clear and well organized since they described both the main IoT technologies and the data alghorithms that can be applied for the scope.

In the following some minor comments:

- Since the topic is related to IoT, if possible, authors can include some more details about "where" alghoritms are applied on data collected by sensors: on board, in the edge, or in the cloud? Is one or more communication protocol needed?

- The english is clear, and I didn't find typos, in my opinion, the text should be revised (specially in sections 2 and 3) in order to uniform notations and acronyms. Some examples:

- the term "WiFi" sometimes appear ad "wifi"

- some acronyms are redefined more timed (e.g. PCA). I suggest authors to check that acronyms are defined only once (in the first occurrence, in Camel Case) and after that only the short version is used

- Only a suggestion: maybe section 2.5.4 can be renamed with "Optical sensor" ?

Author Response

Please see the document attached.

Round 2

Reviewer 1 Report

Comments and Suggestions for Authors

I've gone through the authors' response letter and the new version of
 the document, all my comments have been addressed, and now the paper
 looks much more robust both from a technical and English perspective.
I have just a minor comment regarding Figure 3, which is very
nice, but I think it would be more valuable at the end of Section 1
which would also help in presenting the structure of the paper.

Comments on the Quality of English Language

No comments.

Author Response

Please, check the attached file

Reviewer 2 Report

Comments and Suggestions for Authors

The authors have addressed our comments in the revised version. Therefore, we accept this paper in its current form for publication. 

Author Response

Please, check the attached file
